



# Variational modelling of extreme waves through oblique interaction of solitary waves

Floriane Gidel [1,2] and Onno Bokhove [1]

[1]School of Mathematics, University of Leeds, LS2 9JT Leeds, United Kingdom
[2]Maritime Research Institute Netherlands, 2 Haagsteeg, 6708 PM Wageningen, The Netherlands

*Correspondence to:* Floriane Gidel (mmfg@leeds.ac.uk); Onno Bokhove (O.Bokhove@leeds.ac.uk)

**Abstract.** In this work, we model extreme waves that occur due to Mach reflection through the intersection of two obliquely incident solitary waves. For a given range of incident angles and amplitudes, the Mach stem wave grows linearly in length and amplitude, reaching up to four times the amplitude of the incident waves. A variational approach is used to derive the bidirectional Benney-Luke equations, an asymptotic equivalent of the three-dimensional potential-flow equations modelling water waves. This nonlinear and dispersive model has the advantage of allowing wave propagation in two horizontal directions, which is not the case with the unidirectional Kadomtsev-Petviashvili (KP) equation used in most previous studies. A variational Galerkin finite element method is applied to solve the system numerically in Firedrake with a second-order Stormer-Verlet temporal integration scheme in order to obtain stable simulations that conserve the overall mass and energy of the system. Using this approach, we are able to get close to the fourfold amplitude amplification predicted by Miles.

## 1 Introduction

Offshore structures such as wind turbines, ships and platforms are designed to resist loads and stresses applied by winds, currents and water waves. These three factors can cause damage or destroy these structures when their effect is underestimated. Designers and engineers must take into account the effect of not only each of these phenomena separately but also their interaction, which can increase their adverse effects. In this work, we focus on the impact of extreme waves created from the propagation of an obliquely incident solitary wave along the side of a ship (a wave-structure interaction), or its impact with another identical obliquely incident wave (a wave-wave interaction). These two cases are mathematically equivalent since reflection at a rigid wall (represented here by the ship's side) is modelled through the boundary condition of no normal flow at the wall, which is equivalent to the intersection of two identical waves travelling in opposite directions, in which case a virtual wall is formed. The study of extreme, freak or rogue waves resulting from reflection at a wall or interaction of waves has spawned different theories in the last 50 years, some of which are now reviewed.

The objective of the present work is to apply a theory first introduced in Miles (1977a, b) and based on experiments from Perroud (1957), where he described analytically the behaviour of an incident solitary wave interacting with a wall. For a specific range of angle of incidence $\varphi_i$ and scaled amplitude $a_i$ of the wave, the reflection of the soliton may result in three wave fronts: the incident and reflected waves, as well as a Mach stem wave propagating along the wall with an increasing length (see Fig. 1).





This theory holds in the case of small-but-finite wave's amplitude, shallow-but-finite water depth, and weak nonlinearity, that is

$$\varphi_i^2 \ll O(\epsilon), \qquad a_i = O(\epsilon), \qquad \text{for any} \quad \epsilon \ll O(1), \tag{1}$$

and is based on an interaction parameter, first defined as

$$\kappa = \frac{\varphi_i}{\sqrt{3a_i}}, \tag{2}$$

that enables one to predict the amplitude and direction of propagation of each wave front. The most important observation is the transition at $\kappa = 1$ from a regular reflection ($\kappa \geq 1$) to a Mach reflection ($\kappa < 1$), which has led to the following definition of the stem-wave amplification:

$$\alpha_w = \begin{cases} \dfrac{4}{1 + \sqrt{1 - \kappa^{-2}}}, & \text{for } \kappa \geq 1, \\ (1 + \kappa)^2, & \text{for } \kappa < 1, \end{cases} \tag{3}$$

so that $\alpha_w = a_w / a_i$ is the quotient of the stem wave and incident wave amplitudes. Equation (3) shows that at the transition point where $\kappa = 1$ the stem wave may grow up to four times the amplitude of the incident wave, leading to extreme loading on offshore structures. The aim of the present study is to develop a (numerical) model that can accurately simulate the evolution of the stem wave so that the distance and direction of propagation required to reach the fourfold amplitude can be estimated. A challenging aspect is that it takes a long time and large distance of propagation before the stem wave has reached it maximum amplitude, which was a limit in previous experimental and numerical studies. Kodama et al. (2009) extended Miles' theory to the Kadomtsev-Petviashvili (KP) limit, in which the assumptions are

$$\frac{a_0}{H_0} \ll \epsilon, \quad \left(\frac{H_0}{\lambda_0}\right)^2 \ll \epsilon, \quad \tan^2 \varphi_i \ll \epsilon, \quad \epsilon \ll O(1), \tag{4}$$

where $H_0$, $a_0$ and $\lambda_0$ are the water depth, the wave amplitude and wavelength respectively. While the KP–limit still considers shallow-but-finite depth and small-but-finite amplitudes, the main difference with Miles' theory concerns the condition on the angle $\varphi_i$. Yeh et al. (2010) explained that, contrary to Miles' theory, wherein the soliton propagates in one-direction only (the KdV–limit), the KP–limit assumes a quasi-two dimensional approximation, and therefore the condition $\tan^2 \varphi_i \ll O(\epsilon)$ cannot be simplified to $\varphi_i^2 \ll O(\epsilon)$ as in Miles' assumptions. The quasi-two dimensional KP soliton is not solution of the KdV equation but it can be transformed to an asymptotic KdV soliton via some manipulations detailed in Yeh et al. (2010). However, the width of the obtained KdV soliton is proportional to

$$\sqrt{\frac{a_{KP}}{\cos^2 \varphi_i}}, \tag{5}$$

with $a_{KP}$ the scaled amplitude of the initial KP soliton, and therefore depends on the angle $\varphi_i$. This is physically unrealistic since the KdV soliton should have the same shape whatever its direction of propagation. For this reason, Yeh et al. (2010) brought a "high-order correction" to the solution, setting the amplitude of the KdV soliton to be

$$a_{KdV} = \frac{a_{KP}}{\cos^2 \varphi_i}, \tag{6}$$





so that its width depends on its amplitude $a_{KdV}$, but not on any angle. Taking this into account, they slightly modified the definition (2) of the interaction parameter $\kappa$ to

$$\kappa = \frac{\tan \varphi_i}{\cos \varphi_i \sqrt{3a_i}}, \tag{7}$$

where $a_i = a_{KdV}/H_0$ is the scaled amplitude of the incident wave, leading to what we will hereafter identify as the "modified-
Miles' theory" for the expected stem wave amplification:

$$\alpha_w = \begin{cases} \dfrac{4}{1+\sqrt{1-\kappa^{-2}}}, & \text{for } \kappa \geq 1, \\ (1+\kappa)^2, & \text{for } \kappa < 1, \end{cases} \qquad \text{with} \qquad \kappa = \frac{\tan \varphi_i}{\cos \varphi_i \sqrt{3a_i}}. \tag{3-7}$$

Using this modified interaction parameter in Eq. (3-7), they found much better agreement between previous numerical simulations (Funakoshi, 1980; Tanaka, 1993) and modified-Miles' theory. Moreover, Kodama et al. (2009) showed that the stem wave resulting from the interaction of two solitary waves with small incident angles is an exact solution of the KP equation.
Solving this KP equation, they could describe the exact solution depending on the angle of incidence and the amplitude of the initial waves, and validate their theory with numerical simulations (Kodama et al., 2009; Li et al., 2011). Both the amplitude and length of the stem wave indeed followed their predictions in the case of regular and Mach reflection. The numerical scheme could not simulate the highest amplitudes that Miles predicts for $\kappa \approx 1$. Recently, Ablowitz and Curtis (2013) studied Mach reflection for the Benney-Luke approximation, showing that, in that case, modified-Miles' theory applies asymptotically,
leading to amplifications of up to 3.9.

The purpose of the present work is to derive and apply a stable numerical scheme able to estimate the solution over a long distance of propagation, in order to model high-amplitude waves and to confirm the transition from regular to Mach reflection happening for $\kappa \approx 1$. We develop a model similar to the one of Benney and Luke (1964), which is an asymptotic approximation of the potential-flow equations for small-amplitude and long waves. Whilst it has the advantage of conserving both the nonlinear
and dispersive properties of the waves (essential to the modelling of a freak wave, for instance), it does not require a mesh moving vertically with the free surface since the model is reduced to the horizontal plane. Pego and Quintero (1999) derived these modified Benney-Luke equations and Bokhove and Kalogirou (2016) recently used them to simulate a soliton splash resulting from a wave running in a restricted channel. Their simulations were in reasonable good agreement with experiments, which confirms that the Benney-Luke approximation is an accurate model of water waves. The variational technique used in
the present approach enables to express the equations as a Hamiltonian system on which robust time integrators can be applied (Gagarina et al., 2016). The space and time Galerkin finite element method used to discretise the present model ensures the overall conservation of mass, energy and momentum, which are essential in the high-amplitude and long-distance propagating waves studied here.

The remainder of this paper is organised as follows: the modified Benney-Luke type model is derived from the variational
principle for an inviscid and incompressible fluid (Luke, 1967) in the potential flow approximation, using the small-amplitude and small-dispersion scaling of Pego and Quintero (1999). In order to apply modified-Miles' theory and verify our numerical results against Kodama's exact solution, the KP limit is obtained from the Benney-Luke approximation, leading to a new variational principle for KP. A careful scaling is then defined to obtain an asymptotic soliton solution of our present model, based





on the exact solution of the KP equation from Kodama et al. (2009). The corresponding interaction parameter is consequently derived, leading to another version of modified-Miles' theory (3-7), later used to compare our numerical simulations with respect to Miles' expectations. The present approaches are necessary to determine how to impose the line-solitons on the wave

makers to generate a fourfold amplified wave in the middle of a wave basin and measure its impact on offshore structures. The finite element method is then used to discretise the equations in space together with the second-order Störmer-Verlet temporal scheme that ensures stable simulations. Results are finally discussed and compared to the expectations.

## 2 Water-wave model

### 2.1 Introduction

Our water-wave model is derived from a variational approach that ensures conservation of mass, momentum and energy. In a basic sea state with extreme waves, these conservation properties are essential given the different length scales involved. Starting from Luke's variational principle for an inviscid fluid with a free surface (Luke, 1967), a model similar to the one derived by Benney and Luke (1964) for small-amplitude and long waves is obtained. The (numerical) method developed by Bokhove and Kalogirou (2016) is used to derive the relevant variational principle for our Benney-Luke model. This asymptotic

model conserves the non-linear and dispersive properties of the sea waves, which enables comparison with the Kadomtsev-Petviashvili's (KP) model for which the modified Miles' theory Eq. (3-7) applies.

### 2.2 From Luke's variational principle to Benney-Luke

Water-wave equations are often adequately described by the potential-flow approximation. In the absence of vorticity, the fluid velocity $\mathbf{u} = (u_x, u_y, u_z)$ can be expressed as the gradient of the so-called potential $\phi(x, y, z)$, such that $\mathbf{u} = \nabla \phi$. The deviation

from the surface at rest $H_0$ is defined by $\eta(x, y, t)$ so that the total depth $h(x, y, t)$ can be expressed as $h(x, y, t) = H_0 + \eta(x, y, t)$ (cf. Fig. 2). We consider a flat sea bed lying at $z = 0$, with vertical walls at $\partial \Omega_b$, where $\Omega_b$ is the horizontal plane of the bed coordinates $\Omega_b = \{0 \le x \le L_x, 0 \le y \le L_y\}$. Luke (1967) described an inviscid and incompressible fluid with a free surface in the potential approximation through the following variational principle:

$$\int_0^T \int_{\Omega_b} \int_0^{H_0 + \eta(x,y,t)} \left[ \partial_t \phi + \frac{1}{2} |\nabla_b \phi|^2 + \frac{1}{2} (\partial_z \phi)^2 + g(z - H_0) \right] dz \, dx \, dy \, dt, \tag{8}$$

where $g$ is the acceleration of gravity. The subscript $b$ denotes the horizontal plane of the bed coordinates such that $\nabla_b = (\partial_x, \partial_y)^T$ is the horizontal gradient. The velocity at the walls and sea bed are assumed to be zero, that is $\mathbf{n_b} \cdot \nabla_b \phi = 0$ on $\partial \Omega_b$, with $\mathbf{n_b}$ the outward horizontal normal, and $\partial_z \phi = 0$ at $z = 0$. The boundary conditions at the free surface $z = h$ and the





equations of motion in the domain $\Omega$ are obtained from Eq. (8) as

$$\nabla_b^2 \phi + \partial_{zz}\phi = 0 \qquad\qquad \text{in } \Omega, \tag{9a}$$

$$\partial_t \eta + \nabla\phi \cdot \nabla\eta - \partial_z\phi = 0 \qquad\qquad \text{at } z = h, \tag{9b}$$

$$\partial_t\phi + \frac{1}{2}|\nabla_b\phi|^2 + \frac{1}{2}(\partial_z\phi)^2 + g\eta = 0 \qquad \text{at } z = h, \tag{9c}$$

$$\mathbf{n_b} \cdot \nabla_b\phi = 0 \qquad\qquad \text{on } \partial\Omega_b, \tag{9d}$$

$$\partial_z\phi = 0 \qquad\qquad \text{at } z = 0. \tag{9e}$$

The amplitude parameter $\epsilon = a/H_0 \ll 1$, with $a$ the amplitude of the waves, and the small dispersion parameter $\mu = (H_0/\lambda_0)^2 \ll$ 1, with $\lambda_0$ the horizontal wave length, have been introduced by Pego and Quintero (1999) to scale Eq. (8). The scaled variational principle is

$$0 = \delta \int\limits_0^T \int\limits_{\Omega_b} \int\limits_0^{1+\epsilon\eta} \left[ \epsilon\partial_{\hat{t}}\phi + \frac{\epsilon^2}{2}|\nabla_b\phi|^2 + \frac{1}{2}\frac{\epsilon^2}{\mu}(\partial_{\hat{z}}\phi)^2 \right] d\hat{z} + \frac{1}{2}\epsilon^2\eta^2 \, d\hat{x}\,d\hat{y}\,d\hat{t}, \tag{10}$$

where

$$\hat{x} = \frac{\sqrt{\mu}}{H_0}x, \qquad \hat{y} = \frac{\sqrt{\mu}}{H_0}y, \qquad \hat{z} = \frac{1}{H_0}z, \qquad \hat{t} = \frac{\sqrt{gH_0\mu}}{H_0}t, \qquad \hat{\eta} = \frac{1}{\epsilon H_0}\eta, \qquad \hat{\phi} = \frac{\sqrt{\mu}}{\epsilon H_0\sqrt{\epsilon H_0}}\phi. \tag{11}$$

This scaling focusses on small-amplitude long waves.

To derive the Benney-Luke model, the potential flow $\phi$ is expanded in terms of the sea-bed potential $\phi(x,y,0,t) = \Phi(x,y,t)$ and the dispersion parameter $\mu$, as in Bokhove and Kalogirou (2016):

$$\phi(x,y,z,t) = \Phi(x,y,t) + \mu\Phi_1(x,y,z,t) + \mu^2\Phi_2(x,y,z,t) + \cdots. \tag{12}$$

Combining this expansion with the system of equations (9) and retaining terms up to second order, Eq. (12) becomes (see Bokhove and Kalogirou (2016) for details)

$$\phi = \Phi - \frac{\mu}{2}z^2\Delta\Phi + \frac{\mu^2}{24}z^2\Delta^2\Phi + O(\mu^3). \tag{13}$$

Substituting Eq. (13) into the variational principle (10), one gets the variational principle under the Benney-Luke approximation

$$0 = \delta \int\limits_0^T \int\limits_{\Omega_b} \left[ \eta\partial_t\Phi + \frac{\mu}{2}\nabla\eta \cdot \partial_t\nabla\Phi + \frac{1}{2}(1+\epsilon\eta)|\nabla\Phi|^2 + \frac{\mu}{3}(\Delta\Phi)^2 + \frac{1}{2}\eta^2 \right] dx\,dy\,dt. \tag{14}$$

Arbitrary variations in both $\Phi$ and $\eta$ together with boundary conditions $\mathbf{n}\cdot\nabla\Phi = 0$ and $\mathbf{n}\cdot\Delta\nabla\Phi = 0$ at $\partial\Omega_b$ with $\mathbf{n}$ the normal pointing outward, lead to the Benney-Luke equations

$$\delta\eta: \quad \partial_t\Phi - \frac{\mu}{2}\partial_t\Delta\Phi + \frac{\epsilon}{2}|\nabla\Phi|^2 + \eta = 0, \tag{15a}$$

$$\delta\Phi: \quad \partial_t\eta - \frac{\mu}{2}\partial_t\Delta\eta + \nabla\cdot((1+\epsilon\eta)\nabla\Phi) - \frac{2}{3}\mu\Delta^2\Phi = 0. \tag{15b}$$





These equations will be solved numerically as explained in Sec. 4. However, to test our Benney-Luke model on modified Miles' theory (3-7), it must first be compared to the KP theory for which Kodama et al. (2009) have shown that modified Miles' theory holds.

## 2.3 From Benney-Luke to Kadomtsev-Petviashvili

The Kadomtsev-Petviashvili equation for small-amplitude solitons can be derived from the Benney-Luke variational principle (14) and Eqs. (15) through the transformations

$$X = \sqrt{\frac{\epsilon}{\mu}}(x - t), \qquad Y = \frac{\epsilon}{\sqrt{\mu}}y, \qquad \tau = \epsilon\sqrt{\frac{\epsilon}{\mu}}t, \qquad \Psi = \sqrt{\frac{\epsilon}{\mu}}\Phi, \qquad \eta = \eta. \tag{16}$$

Substituting scalings (16) into Eq. (15a), $\eta$ can be expressed from $\Psi$ as

$$\eta = \Psi_X - \epsilon\Psi_\tau - \frac{\epsilon}{2}\Psi_{XXX} - \frac{\epsilon}{2}(\Psi_X)^2 - \frac{\epsilon^2}{2}(\Psi_y)^2 + \frac{\epsilon^2}{2}\Psi_{\tau XX} - \frac{\epsilon^3}{2}\Psi_{XYY} + \frac{\epsilon^3}{2}\Psi_{\tau YY}. \tag{17}$$

Substituting Eq. (16) into the transformed variational principle (14) yields

$$0 = \delta\int_0^T\int\!\!\int_{\Omega_b}\left[\eta(\epsilon\Psi_\tau - \Psi_X) + \frac{\epsilon}{2}\eta_X(\epsilon\Psi_{\tau X} - \Psi_{XX}) + \frac{\epsilon^2}{2}\eta_Y(\epsilon\Psi_{\tau Y} - \Psi_{XY})\right.$$
$$\left. + \frac{1}{2}(1 + \epsilon\eta)\left((\Psi_X)^2 + \epsilon(\Psi_Y)^2\right) + \frac{\epsilon}{3}\left((\Psi_{XX})^2 + \epsilon^2(\Psi_{YY})^2\right) + \frac{1}{2}\eta^2\right]dX\,dY\,d\tau. \tag{18}$$

Subsequent elimination of $\eta$ using Eq. (17) and truncation to $O(\epsilon^2)$ gives the variational principle for KP in terms of $\eta \approx \Psi_X$:

$$0 = \epsilon\delta\int_0^T\int\!\!\int_{\Omega_b}\left[\Psi_X\Psi_\tau + \frac{1}{2}(\Psi_X)^3 - \frac{1}{6}(\Psi_{XX})^2 + \frac{1}{2}(\Psi_Y)^2\right]dX\,dY\,d\tau \tag{19a}$$

$$= \epsilon\int_0^T\int\!\!\int_{\Omega_b}\delta\Psi\left[-2\Psi_{X\tau} - 3\Psi_X\Psi_{XX} - \frac{1}{3}\Psi_{XXXX} - \Psi_{YY}\right]dX\,dY\,d\tau. \tag{19b}$$

Note that we consider an infinite plane, with $\Psi$ vanishing at the boundaries $|X, Y| \to \infty$, such that the boundary terms arising from the integrations by part vanish in Eq. (19b). Since $\delta\Psi$ is arbitrary, the variational principle (19) yields the following equation for the leading-order scaled potential $\Psi$:

$$2\Psi_{X\tau} + 3\Psi_X\Psi_{XX} + \frac{1}{3}\Psi_{XXXX} + \Psi_{YY} = 0. \tag{20}$$

From Eq. (17), at leading order in $O(\epsilon)$, $\eta$ can be expressed as $\eta = \Psi_X$ and, therefore, taking the partial derivative of Eq. (20) with respect to $X$ leads to the KP equation for $\eta$:

$$\left[2\eta_\tau + 3\eta\eta_X + \frac{1}{3}\eta_{XXX}\right]_X + \eta_{YY} = 0. \tag{21}$$





The solution of the KP equation (21) is found by substituting the following Ansatz, the form inspired by Yeh et al. (2010) Eq. (9), into (21):

$$\eta(X,Y,\tau) = A\,\mathrm{sech}^2\left[B\left(X + Y\tan\varphi - C\tau\right)\right], \tag{22}$$

where $\varphi$ is the angle of incidence, $A$ is the amplitude of the soliton, and $B$ and $C$ ar coefficients to be determined via direct substitution. The KP soliton is then found to be

$$\eta(X,Y,\tau) = A\,\mathrm{sech}^2\left[\sqrt{\frac{3}{4}A}\left(X + Y\tan\varphi - C\tau\right)\right], \tag{23}$$

with $C = \frac{1}{2}A + \frac{1}{2}\tan^2\varphi$, $B = \sqrt{3A/4}$ and $A$ the prescribed amplitude. Using Eq. (17) at leading order, *i.e.* $\eta = \Psi_X$, the solution for $\Psi$ thus becomes

$$\Psi(X,Y,\tau) = \sqrt{\frac{4}{3}A}\left[\tanh\left(\sqrt{\frac{3}{4}A}\left(X + Y\tan\varphi - C\tau\right)\right) + 1\right]. \tag{24}$$

## 3   Comparison with modified Miles' theory and Kodama's exact solution

### 3.1   Introduction to Kodama's exact solution

Kodama et al. (2009) have studied the reflection pattern for a "symmetric V-shape initial waves consisting of two semi-infinite line solitons with the same amplitude", in a system of coordinates $(\tilde{X}, \tilde{Y}, \tilde{\tau})$ related to our system of coordinates (16) $(X, Y, \tau)$ via

$$\tilde{X} = \left(\frac{3}{\sqrt{2}}\right)^{1/3} X, \qquad \tilde{Y} = \left(\frac{3}{\sqrt{2}}\right)^{2/3} Y, \qquad \tilde{\eta} = \frac{1}{3}\left(\frac{3}{\sqrt{2}}\right)^{4/3}\eta, \qquad \tilde{\tau} = \sqrt{2}\tau. \tag{25}$$

They solved the KP equation

$$[4\tilde{\eta}_{\tilde{\tau}} + 6\tilde{\eta}\tilde{\eta}_{\tilde{X}} + \tilde{\eta}_{\tilde{X}\tilde{X}\tilde{X}}]_{\tilde{X}} + 3\tilde{\eta}_{\tilde{Y}\tilde{Y}} = 0, \tag{26}$$

for which the surface deviation solution $\tilde{\eta}$ is given by

$$\tilde{\eta} = \tilde{A}\,\mathrm{sech}^2\left[\sqrt{\frac{\tilde{A}}{2}}\left(\tilde{X} + \tilde{Y}\tan\tilde{\varphi} - \tilde{C}\tilde{\tau}\right)\right], \tag{27}$$

where $\tilde{A}$ is the amplitude of the soliton, $\tilde{\varphi}$ is the angle of incidence at the wall, and $\tilde{C}$ is a constant defined as $\tilde{C} \equiv \frac{1}{2}\tilde{A} +$

$\frac{3}{4}\tan^2\tilde{\varphi}$. They showed that in this specific case, the transition from regular to Mach reflection occurs when

$$\tan\tilde{\varphi} = \sqrt{2\tilde{A}}. \tag{28}$$

Moreover, Kodama et al. (2009) defined exactly the incident, reflected and stem solitons resulting from the interaction as a O-type soliton in the case where $\tan\tilde{\varphi} > \sqrt{2\tilde{A}}$, and a (3142)-type soliton in the case where $\tan\tilde{\varphi} < \sqrt{2\tilde{A}}$. The O–type soliton





consists of two line-solitons travelling in the x–direction, each having a specific amplitude and angle with respect to the y–

axis (see Fig. 3). The $(3142)$–type soliton consists of two other line–solitons, also travelling in the x–direction with their own amplitudes and angles with respect to the y–axis, but this soliton also has the property to be non-stationary, *i.e.* that while it propagates along the x–axis, a new line–soliton is progressively created and grows parallel to the y–axis at the intersection of the two initial line–solitons. In the case of both O-type and (3142)–type solitons, one can indeed associate one of the line–soliton to the incident solitary wave presented in the introduction, the second line–soliton to the reflected wave (with a different amplitude and angle), and the intersection of the two line–solitons as the stem wave, growing in length only when

the angle of the incident wave is smaller than the critical angle (28). These two solitons are represented in Fig. 3, obtained from Kodama et al. (2009). A comparison between these theoretical solitons and those obtained numerically from the V–shape initial soliton showed very good agreement, confirming that the incident, reflected and stem waves described by Miles are indeed asymptotically equivalent to the O–type and (3412)–type solitons, depending on the initial angles. In the case of a symmetric initial pattern, that is for two initial line–solitons of equal amplitude and angle of incidence, Kodama et al. (2009)

gave the expression of the maximal amplitude of the intersection wave, as

$$a_{max} = \begin{cases} \dfrac{1}{2}(\tan\tilde\varphi + \sqrt{2\tilde A})^2 & \text{for } \tan\tilde\varphi < \sqrt{2\tilde A}, \\[2ex] \dfrac{4\tilde A}{(1+\sqrt{1-\dfrac{2\tilde A}{\tan^2\tilde\varphi}})} & \text{for } \tan\tilde\varphi \ge \sqrt{2\tilde A}. \end{cases} \tag{29}$$

Since the condition $\tan\tilde\varphi = \sqrt{2\tilde A}$ is equivalent to Miles' condition $\kappa = 1$, we can define the interaction parameter corresponding to the KP equation (26) as

$$\tilde\kappa = \frac{\tan\tilde\varphi}{\sqrt{2\tilde A}}. \tag{30}$$

Substitution of the interaction parameter (30) into the amplification expectations (29) indeed yields Miles' predictions (3) for $\alpha_w = a_{max}/\tilde A$.

### 3.2 Application to the present Benney-Luke model

In Sec. 2.3, the Benney-Luke model was reduced to the KP equation (21). This equation for the surface deviation $\eta$ is slightly different from the one used by Kodama et al. (2009), and introduced in Eq. (26). In order to compare our numerical solutions

to Kodama et al. (2009)'s results (29)–(30), our KP equation (21) is (re)scaled using the coefficients introduced in Eq. (25), which yields Eq. (26) used by Kodama et al. (2009). Since we know that the KP soliton defined in Eq. (27) is a solution of the KP equation (26), we may transform it back to the initial variables $(X, Y, \tau, \eta)$ introduced in Eq. (16) to get the exact solution of our KP equation (21):

$$\eta = 3\left(\frac{3}{\sqrt 2}\right)^{-4/3} \tilde A \, \text{sech}^2\left[\sqrt{\frac{\tilde A}{2}}\left(\left(\frac{3}{\sqrt 2}\right)^{1/3} X - \tilde C\sqrt 2\tau + \left(\frac{3}{\sqrt 2}\right)^{2/3} Y\tan\tilde\varphi\right)\right]. \tag{31}$$





Hence, the relations between our coefficients $A$, $\varphi$ and $C$ and those appearing in Kodama's solution (27) $\tilde{A}$, $\tilde{\varphi}$ and $\tilde{C}$ are given by

$$A = 3\left(\frac{3}{\sqrt{2}}\right)^{-4/3}\tilde{A}, \qquad C = \left(\frac{4}{3}\right)^{1/3}\tilde{C}, \qquad \tan\varphi = \left(\frac{3}{\sqrt{2}}\right)^{1/3}\tan\tilde{\varphi}, \tag{32}$$

using which the solution (31) becomes

$$\eta = A\operatorname{sech}^2\left[\sqrt{\frac{3}{4}A}\left(X + Y\tan\varphi - C\tau\right)\right], \tag{23}$$

with $C = \frac{1}{2}A + \frac{1}{2}\tan^2\varphi$, which is indeed the solution (23) derived in Sec. 2.3. Therefore, applying scaling (32) to the critical condition (28) yields the critical condition for Eq. (21) and solution (23) as

$$\tan\varphi = \sqrt{3A}. \tag{33}$$

When we transform solution (23) for $\eta$ back to the original Benney-Luke approximation (15) used in our simulations (using scaling (16)), the asymptotic solutions for $\eta$ and $\Psi$ become

$$\eta(x,y,t) = A\operatorname{sech}^2\left[\sqrt{\frac{3\epsilon}{4\mu}A}\left(x - x_0 + \sqrt{\epsilon}(y - y_0)\tan\varphi + (t - t_0)(1 - C\epsilon)\right)\right], \tag{34a}$$

$$\Phi(x,y,t) = \sqrt{\frac{4\mu}{3\epsilon}A}\left[\tanh\left(\sqrt{\frac{3\epsilon}{4\mu}A}\left(x - x_0 + \sqrt{\epsilon}(y - y_0)\tan\varphi + (t - t_0)(1 - C\epsilon)\right)\right) + 1\right], \tag{34b}$$

where the soliton has been localised around the position $(x_0, y_0)$ at time $t = t_0$. Finally, by setting

$$a_i = A, \qquad \tan\varphi_i = \sqrt{\epsilon}\tan\varphi, \qquad \text{and} \qquad \hat{C} = \frac{1}{2}a_i + \frac{1}{2\epsilon}\tan^2\varphi_i, \tag{35}$$

the solutions (34) of the Benney-Luke equations can be rewritten as

$$\eta(x,y,t) = a_i\operatorname{sech}^2\left[\sqrt{\frac{3\epsilon}{4\mu}a_i}\left(x - x_0 + (y - y_0)\tan\varphi_i + (t - t_0)(1 - \hat{C}\epsilon)\right)\right], \tag{36a}$$

$$\Phi(x,y,t) = \sqrt{\frac{4\mu}{3\epsilon}a_i}\left[\tanh\left(\sqrt{\frac{3\epsilon}{4\mu}a_i}\left(x - x_0 + (y - y_0)\tan\varphi_i + (t - t_0)\left(1 - \hat{C}\epsilon\right)\right)\right) + 1\right]. \tag{36b}$$

This solution is used as initial condition at time $t = 0$ in the simulations. Condition (33) defines the following relation between $\varphi_i$, $a_i$ and $\epsilon$ in our Benney-Luke scaling, for Eq. (15):

$$\tan\varphi_i = \sqrt{3\epsilon a_i}. \tag{37}$$

This condition is equivalent to Miles' condition $\kappa = 1$ and therefore we can define our Benney-Luke interaction parameter as

$$\kappa_{BL} = \frac{\tan\varphi_i}{\sqrt{3\epsilon a_i}}. \tag{38}$$





Note however that taking into account the remark from Kodama (2010) about the quasi two-dimensionality of the KP limit, as explained in introduction, the interaction parameter defined in Eq. (38) must be corrected to

$$\kappa_{BL} = \frac{\tan\varphi_i}{\cos\varphi_i \sqrt{3\epsilon a_i}} \tag{39}$$

in order to satisfy Miles' predictions (3). One can see from the potential-flow equations (9) for the Benney-Luke approximation, that the small amplitude parameter $\epsilon$ is defined as $\epsilon = a/h_0$. Therefore, in the specific case where $a_i = 1$ and $\epsilon = a_{KdV}/h_0$, the interaction parameter (7) is recovered. The diagram in Fig. 4 summarizes the equations and solutions derived thus far, in each scaling. In the next section, we explain how the Benney-Luke system of equations are discretized in both space and time in order to be solved numerically.

## 4    Numerical implementation

As a first step in the computational solution, the Benney-Luke model needs to be discretized in space and time, on a meshed domain. This section explains the methods used to discretize the domain and the equations.

### 4.1    Space discretization: Finite Element Method (FEM)

A continuous Galerkin finite element method is used to discretize the solutions in space. The variables $\eta$ and $\phi$ are approximated by the finite element expansion

$$
\begin{aligned}
\eta_h(x,y,t) &= \eta_i(t)\varphi_i(x,y), \\
\Phi_h(x,y,t) &= \Phi_j(t)\varphi_j(x,y),
\end{aligned}
\tag{40}
$$

where the subscript $h$ denotes the discretized form of the solutions with basis functions $\varphi_j(x,y)$, and $i,j \in [1,N]$ with 2N unknowns. The Einstein notation for the implicit summation of repeated indices is used. Substituting expansions (40) into the variational principle (14) yields the space–discretized variational principle

$$0 = \delta \int_0^T \int_{\Omega_b} \left[ \varphi_j \eta_j \varphi_i \dot{\Phi}_i + \frac{\mu}{2}\eta_j \dot{\Phi}_i \nabla\varphi_j \cdot \nabla\varphi_i + \frac{1}{2}(1+\epsilon\varphi_j\eta_j)\Phi_i\Phi_l\nabla\varphi_i \cdot \nabla\varphi_l + \frac{\mu}{3}\Phi_i\Phi_j\Delta\varphi_i\Delta\varphi_j + \frac{1}{2}\varphi_i\varphi_j\eta_i\eta_j \right] d\Omega_b \, dt, \tag{41}$$

with $\dot{\Phi}_i$ the time derivative of $\Phi_i$. Its variation with temporal end-point conditions $\delta\Phi_i(0) = \delta\Phi_i(T) = 0$ is

$$
\begin{aligned}
0 = \int_0^T \delta\Phi_i & \int_{\Omega_b} \left[ -\dot{\eta}_j\varphi_j\varphi_i - \frac{\mu}{2}\dot{\eta}_j\nabla\varphi_j \cdot \nabla\varphi_i + (1+\epsilon\eta_j\varphi_j)\Phi_l\nabla\varphi_i\nabla\varphi_l + \frac{2}{3}\mu\Phi_j\Delta\varphi_j\Delta\varphi_i \right] d\Omega_b \\
& + \delta\eta_i \int_{\Omega_b} \left[ \varphi_i\varphi_j\dot{\Phi}_j + \frac{\mu}{2}\nabla\varphi_i\nabla\varphi_j\dot{\Phi}_j + \frac{\epsilon}{2}\varphi_i\Phi_j\Phi_l\nabla\varphi_j\nabla\varphi_l + \eta_j\varphi_i\varphi_j \right] d\Omega_b \, dt.
\end{aligned}
\tag{42}
$$

To avoid the second-order derivative in the fourth term, the auxiliary variable

$$q(x,y,t) = -\frac{2}{3}\Delta\Phi(x,y,t) \tag{43}$$





is introduced, so that, in the variational principle (14), the term $\frac{\mu}{3}(\Delta\Phi)^2$ can be written as

$$
\begin{aligned}
\frac{\mu}{3}(\Delta\Phi)^2 &= \mu\left(\frac{2}{3}(\Delta\Phi)^2 - \frac{1}{3}(\Delta\Phi)^2\right) \\
&= \mu\left(-\frac{2}{3}\nabla\Delta\Phi\cdot\nabla\Phi - \frac{3}{4}(\frac{2}{3}\Delta\Phi)^2\right) \\
&= \mu\left(\nabla q\cdot\nabla\Phi - \frac{3}{4}q^2\right),
\end{aligned}
\tag{44}
$$

which leads to the variational principle

$$
= \delta\int_0^T\int_{\Omega_b}\left[\eta\partial_t\Phi + \frac{\mu}{2}\nabla\eta\cdot\partial_t\nabla\Phi + \frac{1}{2}(1+\epsilon\eta)|\nabla\Phi|^2 + \mu\left(\nabla q\cdot\nabla\Phi - \frac{3}{4}q^2\right) + \frac{1}{2}\eta^2\right]d\Omega_b\,dt.
\tag{45}
$$

In keeping with Eq. (40), second-order Galerkin expansion for $q$ is now expressed as

$$
q_h(x,y,t) = q_i(t)\varphi_i(x,y).
\tag{46}
$$

Substitution of the expansions (40) and (46) into the variational principle (45) yields the discretized variational principle. Its variations with $\delta\Phi_j(0) = \delta\Phi_j(T) = 0$ lead to the weak formulations in matrix form as in Bokhove and Kalogirou (2016). Rather than using this matrix form directly, we only accommodate the spatial discretization using Firedrake (Rathgeber et al., 2015; Balay et al., 2015, 1997; Dalcin et al., 2011; Hendrickson and Leland, 1995), "an automated system for the portable solution of partial differential equations using the finite element method (FEM)". This automated system uses the finite element method to solve partial differential equations, and requires specification of the following:

– the domain in which the equations are solved, and the kind of mesh to use (*e.g.,* quadilaterals, the spatial dimension, etc.);

– the order and type of polynomials used;

– the type of expansion for the unknowns (*e.g.,* continuous Galerkin, Lagrange polynomials etc.);

– the function space of the unknowns and test functions; and, finally;

– the weak formulations discretized in time.

In the present case, the domain is defined as a horizontal channel ending in an oblique wall, and quadrilaterals are used for its discretization (see details in Sec. 5.1.2). Here, we chose to use quadratic polynomials to expand $\Phi$, $q$ and $\eta$. The resulting weak





formulations implemented in Firedrake in terms of $\Phi_h$, $q_h$ and $\eta_h$ are the following:

$$\delta\Phi_h: \quad 0 = \int_0^T \int_{\Omega_b} \left[ -\partial_t\eta_h\delta\Phi_h - \frac{\mu}{2}\nabla\partial_t\eta_h \cdot \nabla\delta\Phi_h + (1+\epsilon\eta_h)\nabla\delta\Phi_h \cdot \nabla\Phi_h - \mu\nabla q_h \cdot \nabla\delta\Phi_h \right] d\Omega_b\, dt, \tag{47a}$$

$$\delta q_h: \quad 0 = \int_0^T \int_{\Omega_b} \mu\left[ \frac{3}{2}q_h\delta q_h - \nabla\delta q_h \cdot \nabla\Phi_h \right] d\Omega_b\, dt, \tag{47b}$$

$$\delta\eta_h: \quad 0 = \int_0^T \int_{\Omega_b} \left[ \delta\eta_h\partial_t\Phi_h + \frac{\mu}{2}\nabla\delta\eta_h \cdot \nabla\partial_t\Phi_h + \eta_h\delta\eta_h + \frac{\epsilon}{2}\delta\eta_h\nabla\Phi_h \cdot \nabla\Phi_h \right] d\Omega_b\, dt. \tag{47c}$$

The forms given in Eq. (47) are convenient since they highlight the unknowns $\Phi_h$, $q_h$ and $\eta_h$ as well as the test function $\delta\Phi_h$,

$\delta q_h$ and $\delta\eta_h$. The final step is to discretize the equations in time, with a second-order Stormer-Verlet scheme, as explained in the next section.

## 4.2    Time discretization: second-order Stormer-Verlet scheme

The space discretized form of the variational principle (14) can be written in the Hamiltonian form

$$0 = \delta \int_0^T \left[ M_{ij}\Phi_i \frac{d\eta_j}{dt} - H(\Phi_i,\eta_j) \right] dt, \tag{48}$$

where

$$M_{ij} = \int_{\Omega_b} \left[ \varphi_i\varphi_j + \frac{\mu}{2}\nabla\varphi_i \cdot \nabla\varphi_j \right] dx\, dy, \tag{49}$$

and the Hamiltonian

$$H(\phi_i,\eta_j) = \int_{\Omega_b} \left[ \frac{1}{2}(1+\epsilon\eta_j\varphi_j)\Phi_i\Phi_l\nabla\varphi_i \cdot \nabla\varphi_l + \frac{\mu}{3}\Phi_i\Phi_l\Delta\varphi_i \cdot \Delta\varphi_l + \frac{1}{2}\eta_j\eta_l\varphi_j\varphi_l \right] d\Omega_b. \tag{50}$$

Gagarina et al. (2016) have shown that, for a generic Hamiltonian system in the form

$$\delta\mathcal{L}(P,Q,t) = \delta \int_0^T \left( P\frac{dQ}{dt} - H(P,Q) \right) dt, \tag{51}$$

robust time integrators conserving the overall mass and energy can be applied. To derive these time schemes, $P$ and $Q$ are discretized on each time interval $[t^n, t^{n+1}]$ as the approximated momentum $P^\tau$ and coordinate $Q^\tau$, and expanded with coefficients $P^i$ and $Q^i$ and linear continuous basis functions $\varphi^i$ and $\psi^i$:

$$P^\tau = Q^i\varphi^i(t), \qquad Q^\tau = Q^i\psi^i(t). \tag{52}$$





The linear basis functions $\varphi^i$ and $\psi^i$ are continuous within each time interval, but admit discontinuities at the interface between two time slots. Therefore, to discretize Eq. (51), the notion of jumps $[[.]]$ and average $\{\{.\}\}_\alpha^\beta$ for a time dependent function $d(t)$ must be introduced (Gagarina et al., 2016):

$$[[d]]|_{t_n} = d^{n,-} - d^{n,+}, \qquad \text{and} \qquad \{\{d\}\}_\alpha^\beta|_{t_n} = \alpha d^{n,-} + \beta d^{n,+}. \tag{53}$$

The coefficients $\alpha$ and $\beta$ are real numbers defined such that $\alpha + \beta = 1$ and $\alpha, \beta \geq 0$. The notation $d^{n,\pm}$ denotes the left and right traces of $d(t)$ at time $t_n$, that is

$$d^{n,\pm} = \lim_{\epsilon \to 0} d(t_n \pm \epsilon). \tag{54}$$

Discretization of the variational principle Eq. (51) then yields (Gagarina et al., 2016)

$$\delta\mathcal{L}^\tau(P^\tau, Q^\tau, t) = \delta\left[ \sum_{n=0}^{N-1} \int_{t_n}^{t_{n+1}} \left( P^\tau \frac{dQ^\tau}{dt} - H(Q^\tau, P^\tau) \right) dt - \sum_{n=-1}^{N-1} [[Q^\tau]]\{\{P^\tau\}\}_\alpha^\beta|_{t_{n+1}} \right], \tag{55}$$

where N is the number of finite time intervals $[t_n, t_{n+1}]$ that divide the time domain $[0, T]$. Gagarina et al. (2016) showed that to obain a second-order Stormer-Verlet scheme, $P$ and $Q$ must be discretized with a trapezoidal and mid-point rules respectively, that is:

$$P^\tau = \frac{t_{n+1} - t}{\Delta t} P^{n,+} + \frac{t - t^n}{\Delta t} P^{n+1,-}, \tag{56}$$

$$Q^\tau = \frac{2(t - t^n)}{\Delta t} Q^{n+1/2} + \frac{t^n + t^{n+1} - 2t}{\Delta t} Q^n, \tag{57}$$

Substituting Eq. (56-57) into the discretized variational principle (55) yields (Gagarina et al., 2016)

$$\delta\mathcal{L}^\tau(P^\tau, Q^\tau, t) = \delta\left[ \sum_{n=0}^{N-1} \left( (P^{n,+} + P^{n+1,-})(Q^{n+1/2} - Q^{n,+}) - \frac{\Delta t}{2}\left( H(P^{n,+}, Q^{n+1/2}) + H(P^{n+1,-}, Q^{n+1/2}) \right) \right) \right.$$
$$\left. - \sum_{n=-1}^{N-1} (2Q^{n+1/2} - Q^{n,+} - Q^{n+1,+})(\alpha P^{n+1,-} + \beta P^{n+1,+}) \right]. \tag{58}$$

Its variations with end-point conditions $\delta(2Q^{-1/2} - Q^{-1,+}) := \delta Q^{0,-} = 0$ and $\delta P^{0,-} = \delta Q^{N,+} = \delta P^{N,+} = 0$, and conditions $[[P]]|_{t_n} = 0$ and $Q^n = \alpha Q^{n,+} + \beta Q^{n,+}$ with $\alpha \in [0.5, 1]$ and $\beta = 1 - \alpha$ (Gagarina et al., 2016), yields the following second-order Stormer-Verlet scheme:

$$P^{n+1/2} = P^n - \frac{\Delta t}{2} \frac{\partial H(P^{n+1/2}, Q^n)}{\partial Q^n}, \tag{59a}$$

$$Q^{n+1} = Q^n + \frac{\Delta t}{2}\left( \frac{\partial H(P^{n+1/2}, Q^n)}{\partial P^{n+1/2}} + \frac{\partial H(P^{n+1/2}, Q^{n+1})}{\partial P^{n+1/2}} \right), \tag{59b}$$

$$P^{n+1} = P^{n+1/2} - \frac{\Delta t}{2} \frac{\partial H(P^{n+1/2}, Q^{n+1})}{\partial Q^{n+1}}, \tag{59c}$$





with the stability condition

$$|\omega\Delta t| \leq 2, \tag{60}$$

with $\omega$ the waves' frequency. Setting the vectors $P = \{\Phi_i\}$ and $Q = \{\eta_j\}$, the variational principle (48) for Benney-Luke equations can therefore be discretized as in (59), leading to Eq. (A1) in Appendix A. Since the space discetization is performed internally within Firedrake, the weak formulations (A1) can be implemented with the full form of the variables $\Phi_h$ and $\eta_h$ and test functions $\delta\Phi_h$ and $\delta\eta_h$ yielding Eq. (A2), in Appendix A. Substituting the auxiliary variable $q$ defined in Eq. (43), the system of equations (A2) corresponds to the time discretization of Eq. (47), namely

$$0 = \int_{\Omega_b} \left( \Phi_h^{n+1/2} - \Phi_h^n \right) \delta\eta_h + \frac{\mu}{2}\nabla\delta\eta_h \cdot \nabla\left( \Phi_h^{n+1/2} - \Phi_h^n \right) + \frac{\Delta t}{2}\left[ \eta_h^n\delta\eta_h + \frac{\epsilon}{2}\delta\eta_h\nabla\Phi_h^{n+1/2} \cdot \nabla\Phi_h^{n+1/2} \right] d\Omega_b, \tag{61a}$$

$$0 = \int_{\Omega_b} \left( q_h^{n+1/2}\delta q_h - \frac{2}{3}\nabla\delta q_h \cdot \nabla\Phi_h^{n+1/2} \right) d\Omega_b \tag{61b}$$

$$0 = \int_{\Omega_b} \left( \eta_h^{n+1} - \eta_h^n \right) \delta\Phi_h + \frac{\mu}{2}\nabla\delta\Phi_h \cdot \nabla\left( \eta_h^{n+1} - \eta_h^n \right) - \frac{\Delta t}{2}\Bigg[ \left( (1+\epsilon\eta_h^n)\nabla\delta\Phi_h \cdot \nabla\Phi_h^{n+1/2} - \mu\nabla q_h^{n+1/2} \cdot \nabla\delta\Phi_h \right)$$

$$+ \left( (1+\epsilon\eta_h^{n+1})\nabla\delta\Phi_h \cdot \nabla\Phi_h^{n+1/2} - \mu\nabla q_h^{n+1/2} \cdot \nabla\delta\Phi_h \right) \Bigg] d\Omega_b, \tag{61c}$$

$$0 = \int_{\Omega_b} \left( \Phi_h^{n+1} - \Phi_h^{n+1/2} \right) \delta\eta_h + \frac{\mu}{2}\nabla\delta\eta_h \cdot \nabla\left( \Phi_h^{n+1} - \Phi_h^{n+1/2} \right) + \frac{\Delta t}{2}\left[ \eta_h^{n+1}\delta\eta_h + \frac{\epsilon}{2}\delta\eta_h\nabla\Phi_h^{n+1/2} \cdot \nabla\Phi_h^{n+1/2} \right] d\Omega_b. \tag{61d}$$

Timesteps (61a), (61b) and (61c) are implicit, while step (61d) is explicit. Although the equations are nonlinear, one can see

that steps (61b), (61c) and (61d) are linear with respect to the unknowns, $q_h^{n+1/2}$, $\eta_h^{n+1}$ and $\Phi_h^{n+1}$ respectively. Therefore, linear solvers are used to solve these three weak formulations, which reduces the running time by assembling the Jacobian matrix only once instead of computing it at each time step. The implementation of these linear and non-linear solvers is straightforward in Firedrake, since functions that solve weak formulations for specific unknown and test functions already exist (Rathgeber et al., 2015; Balay et al., 1997, 2015; Hendrickson and Leland, 1995; Dalcin et al., 2011).

# 5   Numerical results

In this section, the domain is specified and discretized in order to evaluate $\Phi$ and $\eta$ numerically. The numerical evolution of the stem wave's amplitude is compared to the expectations from our modified-Miles theory Eq. (3) and Eq. (39). Finally, the angle of propagation of the reflected and stem waves are measured and compared to the expectations.

## 5.1   Definition of the domain

### 5.1.1   Orientation of the channel

The interaction of two solitary waves can be modelled using either two obliquely intersecting channels, with incident solitons propagating along each channel (see scheme (a) in Fig. 5), or from the reflection of a soliton at a wall with the no-normal





flow condition at the wall (see scheme (b) in Fig. 5). While the first case (a) is more relevant to the theme of this paper, we choose to model the case (b) to reduce the size of the domain by half and thus to reduce the simulation time. Since the cases

(a) and (b) are mathematically equivalent, the results and conclusions obtained with half of the domain will also be valid for the intersection of two oblique channels.

The domain is described by the length of the wall $L_w$, the length of the channel $L_c$, and the angle of incidence $\varphi_i$. The channel needs to be long enough, compared to the wavelength of the incident wave, in order that the boundaries are far enough from the initial soliton to be considered as being at infinity. From Eq. (34), the width of the initial soliton depends on $\sqrt{3\epsilon/4\mu}$, and since $\mu$ is set to 0.02 for every simulation, the width of the soliton varies with $\epsilon$, from 2.5 (when $\epsilon = 0.20$) to 4 (when $\epsilon = 0.12$). We set $L_c = 5$ to leave enough space between the extremities of the soliton and the boundary of the channel for

every case. To allow the stem wave to grow and reach its maximal amplitude, the wall also needs to be long compared to the wavelength. This constraint was a limit in previous numerical and experimental studies (Tanaka, 1993; Li et al., 2011), since it requires robust and stable numerical schemes and large wave basins. We set the wall length to $200 \le L_w \le 600$ depending on the value of $\epsilon$, that is, more than 100 times the incident wave width. When considering half of the domain as represented in Fig. 5b, we can chose to set the wall in the x- or y-direction, in which case the initial soliton must propagate in an oblique

direction and is therefore equivalent to a KP soliton, as defined in Eq. (36), or we can let the initial soliton propagate in the x- or y-direction, in which case the wall is oblique and the expression of the KP-type soliton (36) can be simplified to a KdV-type soliton propagating in the x- (or y-) direction, as

$$\eta(x,y,t) = a_i \operatorname{sech}^2\left[\sqrt{\frac{3\epsilon}{4\mu}}a_i\left(x - x_0 + (t - t_0)\left(1 - \hat{C}\epsilon\right)\right)\right], \tag{62a}$$

$$\Phi(x,y,t) = \sqrt{\frac{4\mu}{3\epsilon}}a_i\left[\tanh\left(\sqrt{\frac{3\epsilon}{4\mu}}a_i\left(x - x_0 + (t - t_0)\left(1 - \hat{C}\epsilon\right)\right)\right) + 1\right]. \tag{62b}$$

The behaviour of the incident and stem waves in the case of an oblique incident soliton (36) and a soliton propagating in the $x$-direction only (62) are compared in Fig. 6. The initial solitons have amplitude $a_i = 1.0$, small amplitude parameter $\epsilon = 0.14$ and small dispersion parameter $\mu = 0.02$. The angle between the direction of propagation of the solitons and the wall is $\varphi_i = \pi/6$ in both cases. The dashed lines represent the evolution of the interpolated amplitude of incident solitons with time. While the initial amplitude was $a_i = 1.0$ in both cases, we observe that both amplitudes first increase before decreasing to an

asymptotic value, slightly smaller than 1.0 ($a_i = 0.93$). This behaviour is not expected for solitons since they should keep a permanent shape. However, we solve here the Benney-Luke equations for which the KP soliton is only an asymptotic (and not exact) solution. We recall that the transformation (16) from the Benney-Luke model to the KP theory is not exact since it requires a trunctation to $O(\epsilon^2)$. In the numerical simulations represented in Fig. 6, $\epsilon = 0.14$ so the condition $\epsilon \ll O(1)$ is respected only asymptotically which might be responsible for this variation of amplitude. One can however see from Fig. 6 that

the incident KP and KdV–type solitons (36) and (62) converge, and that both do so to the same surface deviation, $a_i = 0.93$. This same limit shows that the approximation error from Benney-Luke to the KP soliton is asymptotically the same as from Benney-Luke to KdV. The stem waves (solid lines in Fig. 6) resulting from the interaction of the KP-type (36) and KdV-type (62) initial solitons with the wall evolve in exactly the same way which confirms that the KP-type and KdV-type initial solitons





(36) and (62) give the same results. The small variations in the curves are due to the mesh resolution which is not fine enough to secure a regular amplitude. However, this approximation is sufficiently accurate to provide an estimate of the asymptotic amplitude of the stem wave. Since we have demonstrated that the two types of initial solitons (36) and (62) evolve similarly to give the same results, subsequent simulations will be conducted using only a unidirectional soliton, as defined by Eq. (62), which is a solution of both the KP and KdV equations.

### 5.1.2 Mesh

In order to evaluate $\Phi$ and $\eta$ at an arbitrary time, the domain is discretised using quadrilaterals. This is done using the mesh generator Gmsh (Geuzaine and Remacle, 2009). Since the domain is large, we define a heterogeneous mesh within which areas of higher refinement along the wall, where the solution needs to be more accurate. Moreover, the end of the domain is truncated with a blunt wall instead of the sharp angle, to avoid boundary quadrilaterals having internal angles that are too acute. The final domain comprising different mesh refinements is represented in Fig. 7, in which the insets show the aforementioned refined mesh and right-hand boundary quadrilateral elements.

### 5.2 Amplification of the stem wave

The numerical amplification of the stem wave is compared with the predictions of modified-Miles' theory applied to our Benney-Luke model (3) and (39), namely

$$
\alpha_w = \begin{cases} \dfrac{4}{1+\sqrt{1-\kappa^{-2}}}, & \text{for } \kappa \geq 1, \\ (1+\kappa)^2, & \text{for } \kappa < 1, \end{cases} \tag{3}
$$

with
$$
\kappa = \frac{\tan\varphi_i}{\cos\varphi_i\sqrt{3\epsilon a_i}}. \tag{39}
$$

The interaction parameter defined in Eq. (39) depends on three parameters: the scaled amplitude of the incident soliton $a_i$, its angle of incidence $\varphi_i$, and the small amplitude parameter $\epsilon$. From Miles' theory, a change in these parameters will modify the behaviour of the reflected and stem waves. Figure 8 shows a comparison between predictions (3) and (39) and numerical simulations for the maximal amplification of the stem wave. The amplitude and angle of incidence of the initial soliton are the same for each of the simulations, with values $a_i = 1.0$ for the amplitude and $\varphi_i = 30°$ for the angle of incidence. Only the small-amplitude parameter $\epsilon$ changes in the different cases, taking values from $0.12$ to $0.20$, which leads to different interaction parameters and thus different evolutions of the stem and reflected waves. This is an alternative choice than in the work of Ablowitz and Curtis (2013) where for a specific $\epsilon$, they run simulations with varying amplitude and angle of incidence. This enabled them to show that the small-amplitude parameter $\epsilon$ has only a weak impact on the amplification of the stem wave for $\kappa < 1$ but limits the amplification with a decrease of $O(\epsilon)$ close to the resonant case $\kappa = 1$, leading for instance to a maximal wave amplification of 3.9 when epsilon = 0.1. Despite this asymptotic limitation in the wave amplification, the purpose of the present simulations is to model wave amplification in various sea state, with various depth of water and characteristic wave





heights, and we do so by using different values of $\epsilon$, since we recall that the small-amplitude parameter $\epsilon$ is the quotient

between the characteristic wave height and the water depth. This will allow the industry to test waves' impact on a wider range

of structures, since different structures are used in different sea states. Moreover, the incident wave amplitude varies slightly

when propagating along the basin. This change has a high impact on the predictions, since a small change of order $O(10^{-2})$

in the incident wave amplitude implies a change of order $O(10^{-2})$ in the interaction parameter, which can lead to a prediction

variation of up to $O(10^{-1})$ near the transition case $\kappa \approx 1$ since the expected amplification varies dramatically in this area. The

amplification $a_w/a_i$ is also affected by a change in the incident amplitude $a_i$. It is therefore necessary to use the accurate value

for the incident amplitude. To obtain Fig. 8, we defined the maximal amplification as follows: when the stem wave reaches

its maximal amplitude $a_{w_{max}}$, we measure the amplitude of the incident wave $a_i$ at the same x-position. This new incident

amplitude $a_i$ is used to adjust the interaction parameter, and to compute the amplification of the stem wave $\alpha_w = a_{w_{max}}/a_i$.

The incident channel has a length $L_c = 5$ and the stem wall $200 \leq L_w \leq 600$. The grid refinement is $0.25 \times 0.25$ in the finest

area (*e.g.* at the wall), and $0.4 \times 1.5$ elsewhere. The numerics follow the theoretical curve, but a slight difference between the

present results and those expected from modified-Miles is noticeable. As alluded to beforehand, we assume that this is due

to the fact that the soliton used as an incident wave is an asymptotic but not exact solution of the Benney-Luke equations.

The scaling from Benney-Luke to KP is not exact but asymptotic, with a truncation at second order, which leads to a slight

difference in the final wave amplification. This observation agrees with the conclusions of Ablowitz and Curtis (2013) on

the asymptotic amplification of the stem wave in the case of the Benney-Luke model. The shift is probably also increased

by the mesh resolution that could be optimised to get a better estimate of the incident wave's amplitude and limit the error

caused by its approximation. New simulations with higher mesh resolution are expected to verify the current results. However,

the present Benney-Luke model still predicts very well the evolution of the stem wave amplitude, enabling it to reach up to

3.6 times the initial amplitude. The stem-wave maximal amplification is reached for $\kappa = 0.9733$, marginally smaller than the

$\kappa = 1.0$ predicted by Miles. While the model from Kodama et al. (2009) could predict perfectly the evolution of the stem wave

based on the KP equation, they were unable to reach more than 3.2 times the initial amplitude in their numerical simulations.

### 5.3   Angle of the stem and reflected waves

Miles's theory also predicts different directions of propagation of the stem and reflected waves in the cases of regular and Mach

reflections. While in the first case, characterised by $\kappa \geq 1$, the angle of the reflected wave $\varphi_r$ is expected to be equal to the one

of the incident soliton $\varphi_i$, it should become larger than $\varphi_i$ in the case of Mach reflection, *i.e.* when $\kappa < 1$:

$$\begin{cases} \varphi_r = \varphi_i & \text{for} \quad \kappa \geq 1 \\ \varphi_r > \varphi_i & \text{for} \quad \kappa < 1. \end{cases} \tag{63}$$

Moreover, in the case of regular reflection, the stem wave is expected to propagate along the wall with a constant length, while

for Mach reflection, its length should increase linearly to make a positive angle $\varphi_w$ with the wall:

$$\begin{cases} \varphi_w = 0 & \text{for} \quad \kappa \geq 1 \\ \varphi_w > 0 & \text{for} \quad \kappa < 1. \end{cases} \tag{64}$$



Predictions (63) and (64) are now being checked numerically.

### 5.3.1 Regular reflection

Figure 9 shows numerical results and expectations for the specific case where $\kappa = 1.12 \geq 1$. The wall makes an angle of $30°$

with the direction of propagation of the initial solitary wave, hence $\varphi_i$ equals $30°$. On the bottom-right plot of Fig. 9, one can measure an angle of $60°$ between the reflected and stem waves which means that the angle $\varphi_r$ between the reflected wave and the line perpendicular to the wall is equal to $30°$, that is, equal to $\varphi_i$. This observation holds at any time and therefore the expectations (63) for the reflected waves are satisfied in the case of regular reflection. The stem wave propagates along the wall without increasing in length, and therefore no angle can be measured between the stem wave and the wall: $\varphi_w = 0$, as predicted

in (64) for regular reflection. These results together with Fig. 8 for the amplification of the stem wave confirm modified-Miles' theory in the case $\kappa \geq 1$, for both the reflected and stem waves.

### 5.3.2 Mach reflection

Figure 10 shows numerical results and schematic expectations for the propagation of the reflected and stem wave for $\kappa = 0.58 < 1$. In the bottom right plot, one can first measure the angle between the incident and reflected waves, as represented in

the top right scheme, to check that $\varphi_r$ is larger than $\varphi_i$. The total angle $\varphi_r + \varphi_i$ measures $70°$, with the initial incident angle set to $\varphi_i = 30°$. Therefore, $\varphi_r$ measures $40°$ and is indeed larger than $\varphi_i$, which corresponds to the predictions. The top right scheme of Fig. 10 also shows that the stem wave length should increase linearly to form an angle $\varphi_w$ with the wall. In the bottom right figure, a top view of the numerical results at different times from $t = 0.28$ to $t = 1.12$ highlights the increase of the stem-wave's length as it propagates along the wall. The dashed orange line connects the solutions, confirming that the

wavelength increases in a linear way.

## 6   Conclusions and discussions

The present model Eq. (15) together with the new scaled interaction parameter (39) shows good agreement with the predictions from Miles concerning the amplification of the stem wave and the angles of the reflected and stem waves. One can observe two different regimes in the numerical results, with different behaviours of the waves in the case of Mach and regular reflections.

This confirms the conclusions obtained by Ablowitz and Curtis (2013) concerning the ability of the Benney-Luke model to predict reflection of obliquely incident solitary waves. Presently, our simulations do not allow determination of the exact value of the interaction parameter at the transition from Mach to regular reflection, but currently the maximal amplification is reached at $\kappa = 0.9733$, which is very close to the predicted maximal amplification at $\kappa = 1.0$. The maximal amplification obtained at the moment is $\alpha_w = 3.6$ which is higher than the amplifications obtained with most previous models and experiments (Kodama

et al., 2009; Li et al., 2011; Tanaka, 1993; Funakoshi, 1980), but still slightly lower than the expected 3.9 amplification from Ablowitz and Curtis (2013). This agrees with the conclusion of Ablowitz and Curtis (2013) concerning the impact of $\epsilon$ on the amplification near $\kappa = 1$. While he obtained the maximal amplification $\alpha_w = 3.9$ for $\epsilon = 0.10$, our amplification $\alpha_w = 3.6$ is





obtained for $\epsilon = 0.17$, which is larger than $0.1$ and thus leads to a larger difference with Miles' prediction of $\alpha_w \approx 4$. Moreover, thanks to the robust scheme used to derive and discretise our equations, that ensures stable simulations over the large domain despite the different length scales involved, our present model is the first model able to describe numerically the dynamic development of the stem wave up to such high amplitudes. Previous studies (Kodama et al., 2009; Li et al., 2011; Tanaka, 1993; Funakoshi, 1980) could not attain such high amplifications because of numerical limitations. Ablowitz and Curtis (2013)

obtained the highest numerical amplification $\alpha_w = 3.9$ by considering the final state initialised immediately yet asymptotically using the KP two-line solution. This last approach gives an accurate understanding of the asymptotic maximal amplification of the stem wave with the BL model, but does not describe the development of the stem wave along the wall. The description and understanding of the wave propagation along the wall is however fundamental for application purposes. The present results, although currently limited by the computational time, are therefore a necessary improvement for the application of obliquely

interacting solitary waves in maritime engineering. More advanced simulations should enable determination of the value of $\kappa$ at the transition from Mach to regular reflection, and to reach higher amplification of the stem wave.

      One can point out some limits to the current model. As already concluded in previous studies, the wave needs to propagate over a long distance (relative to its wavelength) in order to reach its maximal amplitude. Consequently, the numerical domain needs to be large, and the mesh fine enough to estimate the waves' crests accurately. This numerical requirement increases the

computational time. One must therefore find a compromise between the accuracy of the simulations and the running time. This constraint is all the more important in that near the transition from Mach to regular reflection a slight change in the incident wave's amplitude modifies dramatically the interaction parameter and consequently the predictions of the stem and reflected waves. One must therefore be careful when analysing the numerical results. For the same reason, simulations for $\kappa \approx 1$ and large amplifications $\alpha_w \approx 4$ are extremely difficult to obtain, since a slight change in the initial settings ($a_i$, $\epsilon$...) modifies

completely the behaviour of the resulting waves. Li et al. (2011) actually conjectured that the transition between Mach and regular reflection in the neighbourhood of $\kappa = 1$ might be gradual and not as abrupt as expected from Miles' predictions (64).

      Finally, one may wonder how likely this solitary waves' reflection is to occur in an open ocean. Interaction of obliquely incident waves on the wall of ships leads to an increasing wave amplitude near the side, sometimes reaching the deck. This phenomenon is called 'green water', and has been studied experimentally and numerically by the Maritime Research Institute

Netherlands (MARIN) to limit the damage caused by the waves on the ships (Buchner et al., 2014). A comparison between our numerical simulations and experiments may be interesting to explore in the future. The present model can also be used to predict the impact of extreme waves, such as freak or rogue waves, on structures. Indeed, when the stem wave reaches more than twice the amplitude of the incident wave, it can then be viewed as a freak wave since it has similar properties in terms of nonlinearity, dispersivity and high amplitude. Table 1 shows the distance needed by the stem wave to reach more than twice

the incident wave's amplitude in different cases (depending on the value of $\epsilon$). For each value of the small-amplitude parameter $\epsilon$, the numerical distance $L_n$ needed to reach at least twice the amplitude of the initial wave has been measured from the simulations. Then, the definition of the small-amplitude parameter $\epsilon = a_0/H_0$ and the choice of a sea state with characteristic wave height $a_0 = 3m$ enables computation of the corresponding water depth $H_0$. The real distance $L_r$ needed by the wave to propagate in this sea state up to twice the characteristic wave height can then be obtained from scaling (11), with formula





$L_r = L_n \times H_0 / \sqrt{\mu}$. The value of the small-dispersion parameter $\mu$ is set to 0.02 as in the results section. Finally, the wavelength $\lambda_0$ can be obtained from the definition of the small-dispersion parameter $\mu = (H_0/\lambda_0)^2$. In a wave tank where waves can be generated from different directions, one can define the angle of propagation and initial profile of two solitary waves from the

5 asymptotically exact solution Eq. (36) of our model Eqs (15), so that their interaction will lead to a stem wave. The evolution of the stem wave can be predicted from the present model, so an offshore structure such as a scaled ship or a wind turbine can be placed at a position where the stem wave will reach more than twice the initial amplitude of the solitary waves. A scaling of 1/10 from values in Table 1 to experiments leads to achievable incident waves'amplitudes and distance of propagation in MARIN's shallow water basin. By knowing the amplitude of the stem wave at a given position, one can estimate the impact of

10 the wave on structures and validate the predictions with such model tests. The model can thus help the industry to design safer offshore structures that can resist extreme waves' impacts.

The present work can also be used as a starting point for the modelling of the interaction of three obliquely incident line-solitons, which should lead to a ninefold-amplified resulting wave that can also be generated in wave tanks. [1]

## 7  Data availability

The Firedrake implementation of our discretisation of the Benny-Luke equations is an example in Firedrake, www.firedrake.org (Bokhove and Kalogirou, 2016). In addition, the expanded program we used to do our simulation is freely available here.

## Appendix A:  Time-discretization of the present Benney-Luke model

The Störmer-Verlet scheme (59) is applied to the variational principle (48) for Benney-Luke, with $P = \{\Phi_i\}$ and $Q = \{\eta_i\}$, leading to:

$$0 = \int_{\Omega_b} \left( \Phi_i^{n+1/2} - \Phi_i^n \right) \left[ \varphi_i \varphi_j + \frac{\mu}{2} \nabla \varphi_i \cdot \nabla \varphi_j \right] + \frac{\Delta t}{2} \left[ \eta_j^n \varphi_j \varphi_l + \frac{\epsilon}{2} \varphi_j \Phi_i^{n+1/2} \Phi_l^{n+1/2} \nabla \varphi_i \cdot \nabla \varphi_l \right] d\Omega_b, \tag{A1a}$$

$$0 = \int_{\Omega_b} \left( \eta_i^{n+1} - \eta_i^n \right) \left[ \varphi_i \varphi_j + \frac{\mu}{2} \nabla \varphi_i \cdot \nabla \varphi_j \right] - \frac{\Delta t}{2} \left[ \left( (1 + \epsilon \eta_j^n \varphi_j) \Phi_l^{n+1/2} \nabla \varphi_i \cdot \nabla \varphi_l + \frac{2}{3} \mu \Phi_l^{n+1/2} \Delta \varphi_i \cdot \Delta \varphi_l \right) \right.$$

$$\left. + \left( (1 + \epsilon \eta_j^{n+1} \varphi_j) \Phi_l^{n+1/2} \nabla \varphi_i \cdot \nabla \varphi_l + \frac{2}{3} \mu \Phi_l^{n+1/2} \Delta \varphi_i \cdot \Delta \varphi_l \right) \right] d\Omega_b, \tag{A1b}$$

$$0 = \int_{\Omega_b} \left( \Phi_i^{n+1} - \Phi_i^{n+1} \right) \left[ \varphi_i \varphi_j + \frac{\mu}{2} \nabla \varphi_i \cdot \nabla \varphi_j \right] + \frac{\Delta t}{2} \left[ \eta_j^{n+1} \varphi_j \varphi_l + \frac{\epsilon}{2} \varphi_j \Phi_i^{n+1/2} \Phi_l^{n+1/2} \nabla \varphi_i \cdot \nabla \varphi_l \right] d\Omega_b. \tag{A1c}$$

---

[1]O. Bokhove suggested this calculation to Prof. Y. Kodama, personal communication, who performed the calculation using the KP equation at the international workshop "Rogue waves" held at the Max Planck Institute in 2011, Dresden, Germany.





Since the space discetization is performed internally within Firedrake, the weak formulations (A1) can be implemented with the full form of the variables $\Phi_h$ and $\eta_h$ and test functions $\delta\Phi_h$ and $\delta\eta_h$ as

$$0 = \int_{\Omega_b} \left(\Phi_h^{n+1/2} - \Phi_h^n\right)\delta\eta_h + \frac{\mu}{2}\nabla\delta\eta_h \cdot \nabla\left(\Phi_h^{n+1/2} - \Phi_h^n\right) + \frac{\Delta t}{2}\left[\eta_h^n\delta\eta_h + \frac{\epsilon}{2}\delta\eta_h\nabla\Phi_h^{n+1/2} \cdot \nabla\Phi_h^{n+1/2}\right]d\Omega_b, \tag{A2a}$$

$$0 = \int_{\Omega_b} \left(\eta_h^{n+1} - \eta_h^n\right)\delta\Phi_h + \frac{\mu}{2}\nabla\delta\Phi_h \cdot \nabla\left(\eta_h^{n+1} - \eta_h^n\right) - \frac{\Delta t}{2}\left[\left((1+\epsilon\eta_h^n)\nabla\delta\Phi_h \cdot \nabla\Phi_h^{n+1/2} + \frac{2}{3}\mu\Delta\delta\Phi_h \cdot \Delta\Phi_h^{n+1/2}\right)\right.$$

$$\left. + \left((1+\epsilon\eta_h^{n+1})\nabla\delta\Phi_h \cdot \nabla\Phi_h^{n+1/2} + \frac{2}{3}\mu\Delta\delta\Phi_h \cdot \Delta\Phi_h^{n+1/2}\right)\right]d\Omega_b, \tag{A2b}$$

$$0 = \int_{\Omega_b} \left(\Phi_h^{n+1} - \Phi_h^{n+1/2}\right)\delta\eta_h + \frac{\mu}{2}\nabla\delta\eta_h \cdot \nabla\left(\Phi_h^{n+1} - \Phi_h^{n+1/2}\right) + \frac{\Delta t}{2}\left[\eta_h^{n+1}\delta\eta_h + \frac{\epsilon}{2}\delta\eta_h\nabla\Phi_h^{n+1/2} \cdot \nabla\Phi_h^{n+1/2}\right]dx\,dy. \tag{A2c}$$

*Competing interests.* The authors declare that they have no conflict of interest.

*Acknowledgements.* The research was funded by the Marie-Curie Fellowship, as part of the European Industry Doctorate (EID) SurfsUp
project. The Firedrake implementation of our discretisation of the Benny-Luke equations is an example in Firedrake, www.firedrake.org;
special thanks to Dr. Anna Kalogirou (Leeds) for sharing this code. In addition, the expanded program we used to do our simulation is
freely available here. We finally express our sincere gratitude to Prof. Mark Kelmanson (Leeds) and Geert Kapsenberg (MARIN) for their
comments, suggestions and corrections that considerably improved this paper.



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





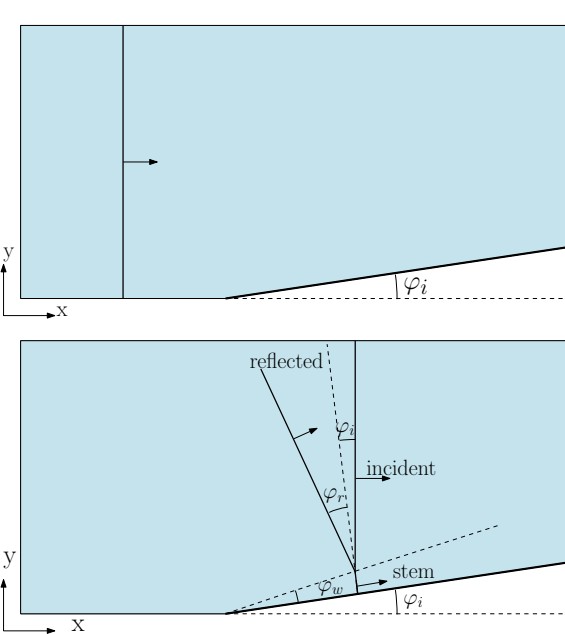

**Figure 1.** Top: top view of a channel containing an incident solitary wave propagating in the $x-$direction with amplitude $a_i$. The side wall is oblique and makes an angle $\varphi_i$ with the $x-$direction. Bottom: top view of the reflection pattern when the incident wave impacts the wall. The pattern is composed of three waves: 1) the incident wave, 2) a reflected wave of amplitude $a_r$ that forms an angle $\varphi_r$ with the angle perpendicular to the wall, and 3) a Mach stem wave propagating along the wall with amplitude $a_w$ and an angle $\varphi_w$ with the wall.





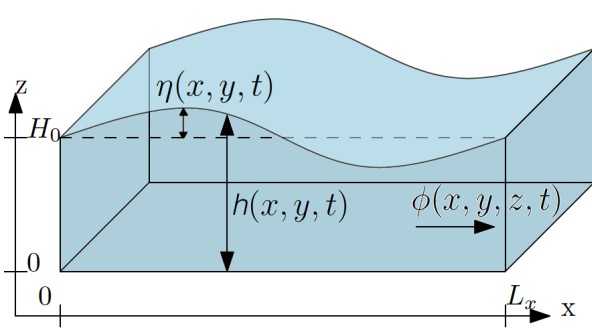

**Figure 2.** Three-dimensional water-wave domain with depth of rest $H_0$. We aim to estimate the potential $\phi(x, y, z, t)$ and the free-surface deviation $\eta(x, y, t)$ from the rest depth.





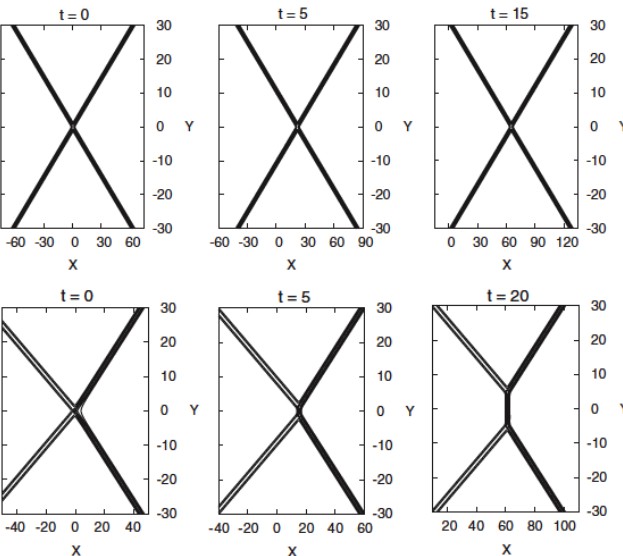

**Figure 3.** O–type and (3142)–type solitons as represented by Kodama et al. (2009). Top: evolution (from left to right) of the O-type soliton, consisting of two line–solitons with different amplitudes and angles with respect to the y–axis. As it propagates, the shape of this soliton is kept unchanged. Bottom: evolution (from left to right) of the (3142)–type soliton, consisting of two line–solitons travelling in the x–direction with different angles and amplitudes. As the soliton propagates, a new line–soliton is created at the intersection of the two initial line–solitons, leading to a stem wave. Figure obtained from Kodama et al. (2009).





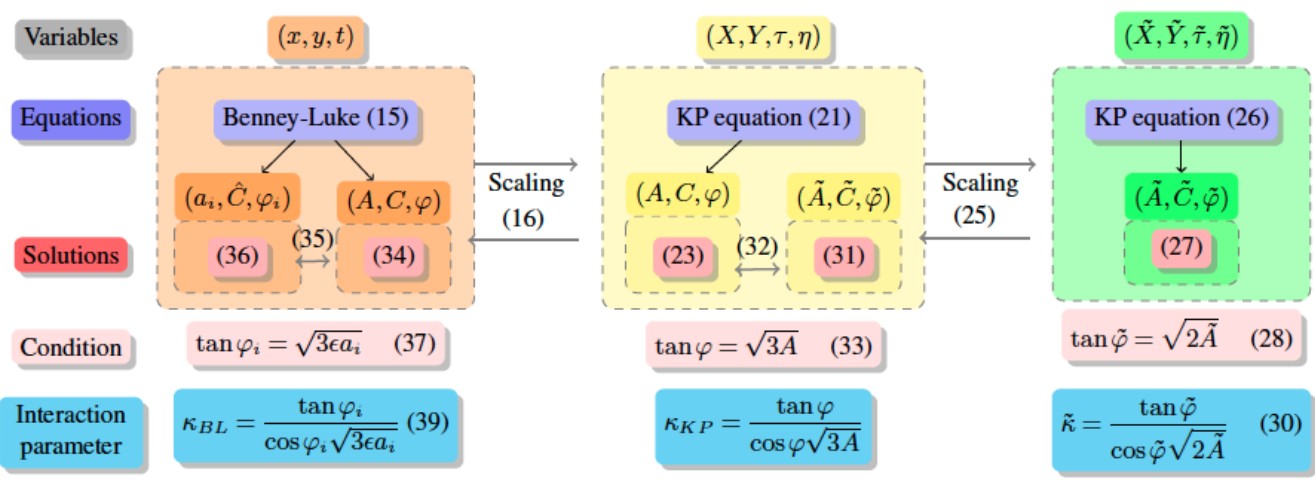

**Figure 4.** Link between the (scaling of the) three system of equations involved in the derivation of the exact solution and critical condition for which Miles and Kodama's predictions hold in the Benney-Luke approximation.

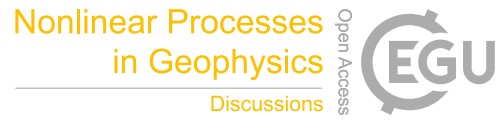



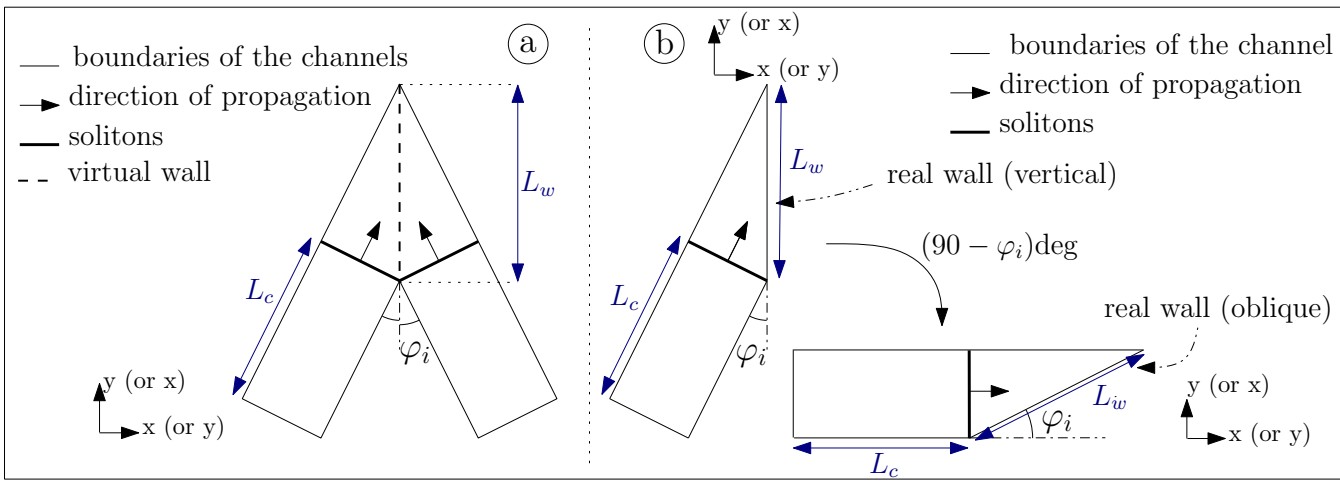

**Figure 5.** Definition of the domain in the two cases described in the text: a) intersection of two channels, with two obliquely incident solitons interacting at a virtual wall, and b) half of the domain with a soliton propagating in one channel and colliding with an oblique wall. This wall is either in the $x-$ or $y-$direction (in which case the soliton has a two-dimensional propagation of direction) or oblique, in which case the incident soliton propagates in a one-dimensional direction ($x$ or $y$).





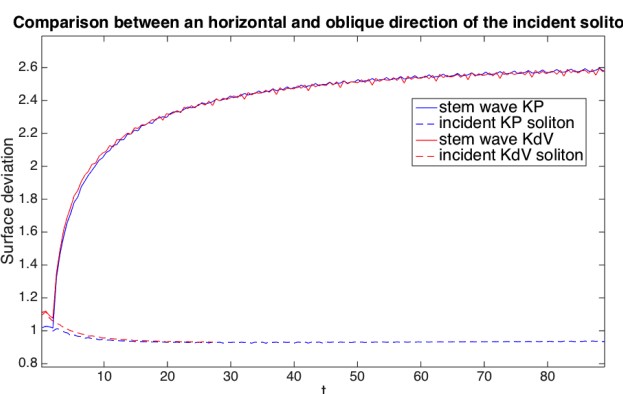

**Figure 6.** Results obtained for an initial amplitude $a_i = 1.0$, and angle $\varphi_i = \pi/6$ rad. Blue: behaviour of the incident and stem wave when the incident soliton propagates in an oblique direction; Red: behaviour of the incident and stem wave when the incident soliton propagates in one direction.





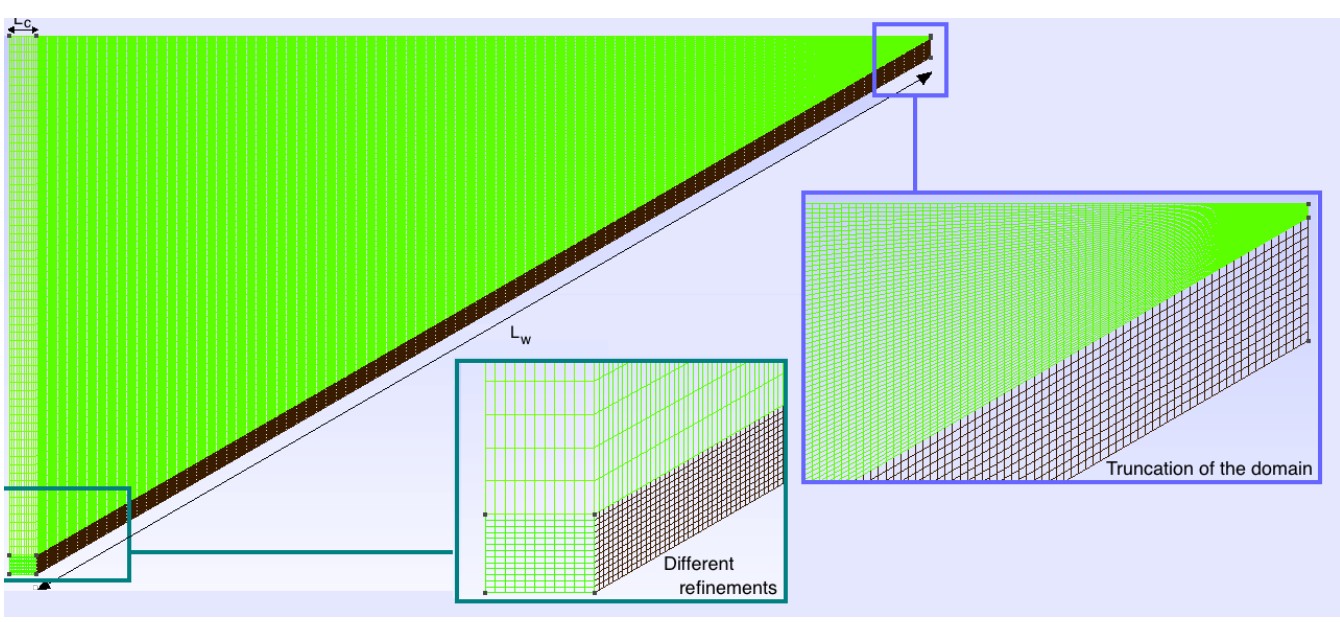

**Figure 7.** Discretised domain with quadrilaterals. The mesh is refined along the wall only, to reduce the computational time.





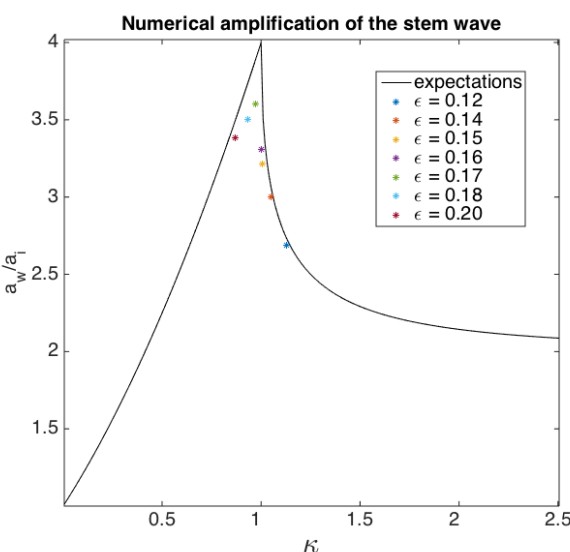

**Figure 8.** Comparison between the expected amplification (solid line) from Miles (3) and our numerical results (symbols) for different values of the interaction parameter $\kappa$, namely: $\kappa \approx 1.1265$ ($\epsilon = 0.12$), $\kappa \approx 1.0526$ ($\epsilon = 0.14$), $\kappa \approx 1.0077$ ($\epsilon = 0.15$), $\kappa \approx 0.9989$ ($\epsilon = 0.16$), $\kappa \approx 0.9733$ ($\epsilon = 0.17$), $\kappa \approx 0.9345$ ($\epsilon = 0.18$), $\kappa \approx 0.8692$ ($\epsilon = 0.20$).





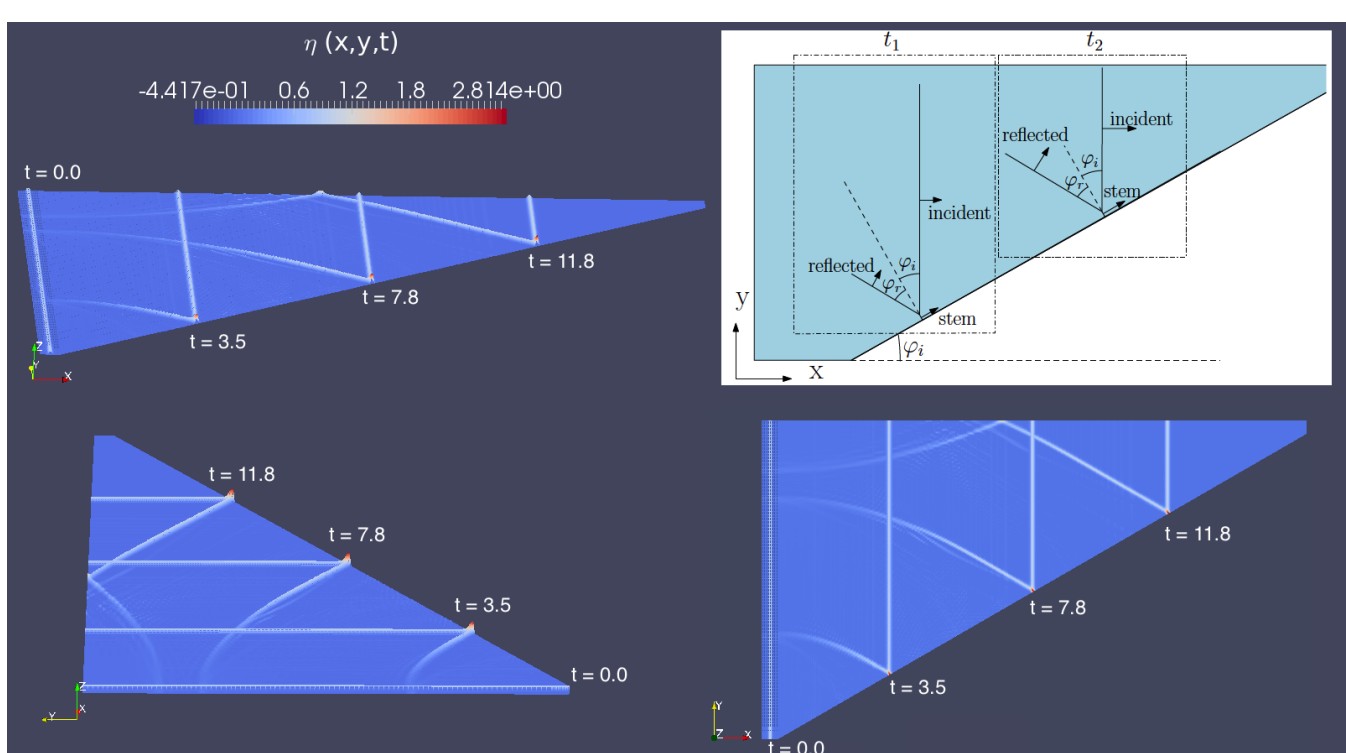

**Figure 9.** Numerical results and expectations for the surface deviation $\eta$ at different times. Left: numerical evolution of the incident, reflected and stem waves from different views. Right: top view of the expected (top) and numerical (bottom) reflections of the incident solitary wave at different times. For $t_1 < t_2$, we expect the angle $\varphi_r$ to be constant and equal to the incident angle $\varphi_i$.





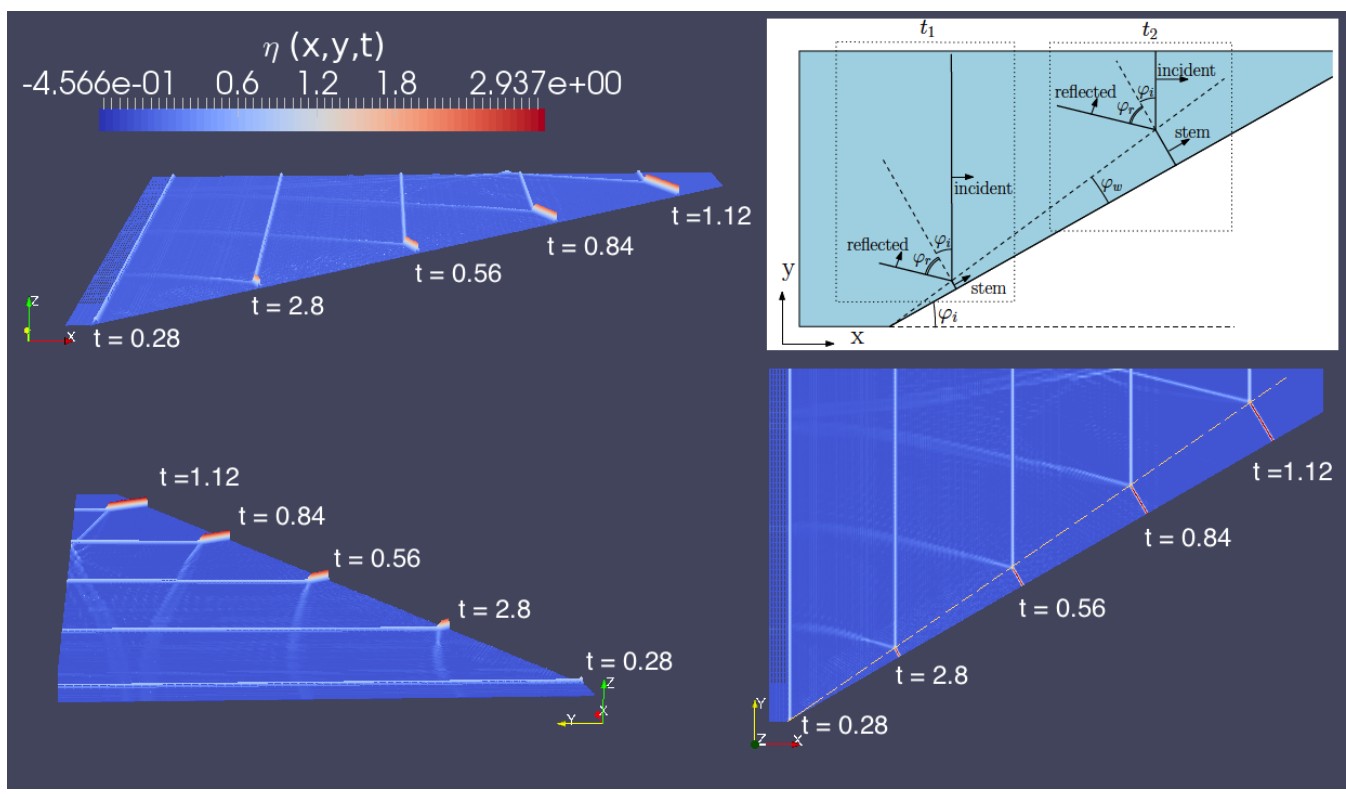

**Figure 10.** Numerical results and expected behaviour of the reflected and stem waves in the case of Mach reflection, that is $\kappa < 1$. Left: numerical evolution of the incident, reflected and stem waves. Top right: top-view scheme of the expected evolution of the stem and reflected waves at two different times $t_1$ and $t_2$ with $t_1 < t_2$. The stem wave should linearly grow in length, leading to an angle $\varphi_w > 0$ rad with the wall. Bottom right: top view of the numerical results. The dashed orange line connects the stem wave lengths at different times, showing that it indeed grows linearly.





| | $\epsilon$ | | | | | | |
| --- | --- | --- | --- | --- | --- | --- | --- |
| | 0.12 | 0.14 | 0.15 | 0.16 | 0.17 | 0.18 | 0.20 |
| Numerical distance $L_n$ | 5.8 | 5.5 | 5.5 | 7.8 | 7.7 | 8.0 | 8.0 |
| Water depth $H_0$ (m) | 25.00 | 21.43 | 20.00 | 18.75 | 17.65 | 16.67 | 15.00 |
| Real distance $L_r$ (m) | 1025 | 833 | 778 | 1028 | 965 | 940 | 846 |
| Wave length $\lambda_0$ (m) | 176.78 | 151.52 | 141.42 | 132.58 | 124.78 | 117.85 | 106.07 |

**Table 1.** Prediction of the minimal distance needed by the stem wave to reach at least twice its initial amplitude in a sea state with characteristic wave's height $a_0 = 3$ m. The dispersion parameter $\mu$ is set to 0.02 while the small-amplitude parameter $\epsilon$ varies from 0.12 to 0.20, leading to different wave evolutions. The numerical distance needed to reach more than twice the incident wave's amplitude is measured from the numerical simulations. The corresponding water depth, real distance of propagation and wavelength are computed from the definition of $\epsilon$, $\mu$, and scaling (11). These values are approximate.