# Peer review of "Variational modelling of extreme waves through oblique interaction of solitary waves: application to Mach reflection"

_Nonlinear Processes in Geophysics, 2016_

## Referee Comment (RC1) · Anonymous Referee #1 · 3 Nov 2016

The paper re-examines the problem of oblique interaction of a plane soliton with a rigid boundary. This problem is known as the Mach reflection and has been studied within the framework of the unidirectional Kadomtsev–Petviashvili (KP) equation. In the reviewed paper a variational approach is used to derive a more general the bidirectional Benney–Luke (BL) set of equations which represent a shallow-water asymptotic equivalent of the three-dimensional set of hydrodynamic equations for water waves. Within the derived BL set equations one can study wave propagation in two horizontal directions, whereas for unidirectional wave propagation the set reduces to the conventional KP equation. The set of equations is solved numerically to obtain a solution for the Mach stem through the intersection of two obliquely incident solitary waves. It is shown that for a given range of incident angles and amplitudes of solitons, the Mach stem grows linearly in length and amplitude, reaching up to four times the amplitude of the

incident solitary waves. Such a big grows of amplitude makes the stem wave a good candidate for the rogue waves on shallow water.

The paper is topical and interesting for a wide audience. It is well written and contains valuable results. It is a pleasure to see it published. I would only advise to make minor stylistical corrections in the section titles:

1) 2.2 From Luke's variational principle to Benney–Luke set of equation; 2) 2.3 From Benney–Luke set of equations to the Kadomtsev-Petviashvili equation.

---

## Referee Comment (RC2) · Anonymous Referee #2 · 5 Nov 2016

This paper is about the reflection of obliquely incident solitary waves on vertical walls. This is a well studied problem and it is known that at small angles a Mach stem is observed which can lead to large amplitudes (up to 4 times the incident wave). The main goal of the paper is to devise a finite element numerical scheme that can be used to solve the Benney-Luke equation - an equation which encompasses previous studies using the KP equation. The paper is well written but probably spends too many pages re-deriving BL and KP (albeit using variational methods). The name Benney-Luke was unfortunately seldom used in the literature (to my knowledge it reappears in Milewski & Keller 1996 and Pego & Quintero 1999) and, although I am not aware of any, I wonder if are any studies of Mach reflection using "three-dimensional Boussineq" (which is essentially what Benney-Luke is) models in the literature.

I would note that recently Kodama and Yeh have claimed that the KP order is insufficient

to capture the large amplitude Mach stem and that better results are obtained using higher order corrections to KP. These would be out of the range of the present BL equation. However, given how BL performs here one may wonder whether this claim is correct.

Note that the "extreme wave" claim for such cross wave constructions in shallow water is not new (see Peterson et al).

I also believe the title is a bit too broad. It should probably mention the oblique reflection of a solitary wave with a wall. (perhaps keep the same title and add ": application to Mach reflection.")

I am happy to recommend this paper for publication. Incidentally I do not find the way the actual waves are presented (Figures 9 & 10) particularly informative - a larger version of the top view (i.e. the bottom right panel) only would be better.

Milewski, P. A., & Keller, J. B. (1996). Three‐Dimensional Water Waves. Studies in Applied Mathematics, 97(2), 149-166.

Pego, R. L., & Quintero, J. R. (1999). Two-dimensional solitary waves for a Benney–Luke equation. Physica D: Nonlinear Phenomena, 132(4), 476-496.

Kodama, Y., & Yeh, H. (2016). The KP theory and Mach reflection. Journal of Fluid Mechanics, 800, 766-786.

Peterson, P., Soomere, T., Engelbrecht, J., & Van Groesen, E. (2003). Soliton interaction as a possible model for extreme waves in shallow water. Nonlinear Processes in Geophysics, 10(6), 503-510.
* * *

---

## Author Comment (AC1) · 21 Nov 2016

Thank you for your comment and suggestions. We agree with your summary and we have accommodated the following section titles changes in our manuscript:

1) 2.2 From Luke's variational principle to Benney–Luke set of equation;

2) 2.3 From Benney–Luke set of equations to the Kadomtsev-Petviashvili equation.

Best regards,

Floriane Gidel and Onno Bokhove

---

## Author Comment (AC2) · 21 Nov 2016

Thank you for commenting our article. Below is the answer point by point to your comments and suggestions:

*"This paper is about the reflection of obliquely incident solitary waves on vertical walls. This is a well studied problem and it is known that at small angles a Mach stem is observed which can lead to large amplitudes (up to 4 times the incident wave). The main goal of the paper is to devise a finite element numerical scheme that can be used to solve the Benney-Luke equation - an equation which encompasses previous studies using the KP equation."*
Thank you for this summary. We agree.

*"The paper is well written but probably spends too many pages re-deriving BL and KP (albeit using variational methods)."*
Given that we require the variational formulation for our numerical techniques we feel our presentation is presently of the correct length to accommodate understanding of our work by the readership. Our variational approach is helpful therein.

*"The name Benney-Luke was unfortunately seldom used in the literature (to my knowledge it reappears in Milewski  Keller 1996 and Pego  Quintero 1999) and, although I am not aware of any, I wonder if are any studies of Mach reflection using "three-dimensional Boussineq" (which is essentially what Benney-Luke is) models in the literature."*
Thank you for this remark; now included the reference to Milewski and Keller (1996).

*"I would note that recently Kodama and Yeh have claimed that the KP order is insufficient to capture the large amplitude Mach stem and that better results are obtained using higher order corrections to KP. These would be out of the range of the present BL equation. However, given how BL performs here one may wonder whether this claim is correct. "*
As we derive our initial BL condition from the KP equation, we actually include a higher order correction to our interaction parameter in Equ. (39), following the remark from Yeh et al (2010). This leads to a much better agreement between our numerical results and the theoretical expectations, which confirms their claim. Is this something different from what you mean in your comment? If so, can you please clarify what you mean by "higher order correction to KP"? Thank you.

*"Note that the "extreme wave" claim for such cross wave constructions in shallow water is not new (see Peterson et al). "*
Indeed, and they also explained in which conditions these waves may occur in real conditions, which may complete our note about 'green water'.  Thank you for this

remark; we have added a reference to this paper.

*"I also believe the title is a bit too broad. It should probably mention the oblique reflection of a solitary wave with a wall. (perhaps keep the same title and add ": application to Mach reflection.")"*
We have extended the title.

*"I am happy to recommend this paper for publication. Incidentally I do not find the way the actual waves are presented (Figures 9 10) particularly informative - a larger version of the top view (i.e. the bottom right panel) only would be better."*
Thank you for this suggestion. We have extended and improved these figures. They now contain a larger version of the top view of the numerical results, as well as a scheme of the expected behaviour of the stem and reflected waves for comparison. We also increased the size of the side view in order to highlight the difference between the stem and incident waves' amplitudes. We have removed the front view, which indeed did not bring any complementary information. You can find these new figures in the attached supplement.

Thank you for all these suggestions. An updated version of the article is available in the attached supplement.
Best regards,
Floriane Gidel and Onno Bokhove

Please also note the supplement to this comment:
http://www.nonlin-processes-geophys-discuss.net/npg-2016-58/npg-2016-58-AC2-supplement.pdf

**Supplement:**

[revised manuscript text omitted]

**Figure 10.** Numerical results and expectations for the reflected and stem waves in the case of Mach reflection, that is $\kappa < 1$. Left: top view of the numerical evolution of the incident, reflected and stem waves. Top right: top view scheme of the expected evolution of the stem and reflected waves at two different times $t_1$ and $t_2$ with $t_1 < t_2$. The stem wave should linearly grow in length, leading to an angle $\varphi_w > 0$ rad with the wall. The angle $\varphi_r$ of the reflected wave is expected to be constant and larger than the incident wave's angle $\varphi_i$. Bottom right: side view of the time evolution of the incident, reflected and stem waves highlighting the amplification of the stem wave's amplitude compared to the initial solitary wave's height.

| | | | | $\epsilon$ | | | |
| --- | --- | --- | --- | --- | --- | --- | --- |
| | 0.12 | 0.14 | 0.15 | 0.16 | 0.17 | 0.18 | 0.20 |
| Numerical distance $L_n$ | 5.8 | 5.5 | 5.5 | 7.8 | 7.7 | 8.0 | 8.0 |
| Water depth $H_0$ (m) | 25.00 | 21.43 | 20.00 | 18.75 | 17.65 | 16.67 | 15.00 |
| Real distance $L_r$ (m) | 1025 | 833 | 778 | 1028 | 965 | 940 | 846 |
| Wave length $\lambda_0$ (m) | 176.78 | 151.52 | 141.42 | 132.58 | 124.78 | 117.85 | 106.07 |

**Table 1.** Prediction of the minimal distance needed by the stem wave to reach at least twice its initial amplitude in a sea state with characteristic wave's height $a_0 = 3$ m. The dispersion parameter $\mu$ is set to $0.02$ while the small-amplitude parameter $\epsilon$ varies from $0.12$ to $0.20$, leading to different wave evolutions. The numerical distance needed to reach more than twice the incident wave's amptitude is measured from the numerical simulations. The corresponding water depth, real distance of propagation and wavelength are computed from the definition of $\epsilon$, $\mu$, and scaling (11). These values are approximate.

---

## Author Response (AR1)

**Final Author's Response to Reviewers #1 and #2 for the article "Variational modelling of extreme waves through oblique interaction of solitary waves: application to Mach reflection"**

**I. Response to Reviewer #1's comments**

**Reviewer's comments:**

The paper re-examines the problem of oblique interaction of a plane soliton with a rigid boundary. This problem is known as the Mach reflection and has been studied within the framework of the unidirectional Kadomtsev–Petviashvili (KP) equation. In the reviewed paper a variational approach is used to derive a more general the bidirectional Benney–Luke (BL) set of equations which represent a shallow-water asymptotic equivalent of the three-dimensional set of hydrodynamic equations for water waves. Within the derived BL set equations one can study wave propagation in two horizontal direc-tions, whereas for unidirectional wave propagation the set reduces to the conventional KP equation. The set of equations is solved numerically to obtain a solution for the Mach stem through the intersection of two obliquely incident solitary waves. It is shown that for a given range of incident angles and amplitudes of solitons, the Mach stem grows linearly in length and amplitude, reaching up to four times the amplitude of the incident solitary waves. Such a big grows of amplitude makes the stem wave a good candidate for the rogue waves on shallow water.

The paper is topical and interesting for a wide audience. It is well written and contains valuable results. It is a pleasure to see it published. I would only advise to make minor stylistical corrections in the section titles:

1) 2.2 From Luke's variational principle to Benney–Luke set of equation;

2) 2.3 From Benney–Luke set of equations to the Kadomtsev-Petviashvili equation.

**Author's response:**

Thank you for your helpful comments and suggestions. We have accommodated the following section-title changes in our manuscript:

2.2 From Luke's variational principle to Benney–Luke set of equations;
 2.3 From Benney–Luke set of equations to the Kadomtsev-Petviashvili equation.

**II. Response to Reviewer #2's comments**

• **Reviewer's comment:** This paper is about the reflection of obliquely incident solitary waves on vertical walls. This is a well studied problem and it is known that at small angles a Mach stem is observed which can lead to large amplitudes (up to 4 times the incident wave). The main goal of the paper is to devise a finite element numerical scheme that can be used to solve the Benney-Luke equation - an equation which encompasses previous studies using the KP equation.

**Author's response:** Thank you for commenting our article. Below is the point-by-point response to your helpful comments and suggestions.

• **Reviewer's comment:** The paper is well written but probably spends too many pages re-deriving BL and KP (albeit using variational methods).

**Author's response:** Given that we require a variational formulation underpins our numerical technique, we feel that our presentation is currently of the correct length to facilitate understanding of our work by the readership.

• **Reviewer's comment:** The name Benney-Luke was unfortunately seldom used in the literature (to my knowledge it reappears in Milewski & Keller 1996 and Pego & Quintero 1999) and, although I am not aware of any, I wonder if are any studies of Mach reflection using "three-dimensional Boussineq" (which is essentially what Benney-Luke is) models in the literature.

**Author's response:** Thank you for this remark; we have now inserted the reference to Milewski and Keller (1996).

• **Reviewer's comment:** I would note that recently Kodama and Yeh have claimed that the KP order is insufficient to capture the large amplitude Mach stem and that better results are obtained using higher order corrections to KP. These would be out of the range of the present BL equation. However, given how BL performs here one may wonder whether this claim is correct.

**Author's response:** As we derive our initial BL condition from the KP equation, we actually include a higher order correction to our interaction parameter in Eq.(39), following the remark from Yeh et al. (2010). This leads to a much better agreement between our numerical results and the theoretical expectations, which confirms Kodama and Yeh's claim.

• **Reviewer's comment:** Note that the "extreme wave" claim for such cross wave constructions in shallow water is not new (see Peterson et al).

**Author's response:** Indeed, and in the paper you refer to, Peterson et al. also explained under which conditions extreme waves may occur in real conditions, which may complete our note about 'green water'. Thank you for this remark; we have added this reference to our paper.

• **Reviewer's comment:** I also believe the title is a bit too broad. It should probably mention the oblique reflection of a solitary wave with a wall. (perhaps keep the same title and add ": application to Mach reflection.")

Author's response: We have extended the title.

• **Reviewer's comment:** I am happy to recommend this paper for publication. Incidentally I do not find the way the actual waves are presented (Figures 9 & 10) particularly informative - a larger version of the top view (i.e. the bottom right panel) only would be better.

**Author's response:** Thank you for this suggestion. We have extended and improved these figures. They now contain a larger version of the top view of the numerical results, as well as a scheme of the expected behaviour of the stem and reflected waves for comparison. We also increased the size of the side view in order to highlight the difference between the stem and incident waves' amplitudes. We have removed the front view, which indeed did not add any complementary information.

**III.** Changes in the manuscript**

Changes accruing from the Reviewers' comments:

- 1) We have accommodated changes in sections 2.2 and 2.3's titles;
- 2) We have added references to the following papers:

- a. Milewski, P. and Keller, J.: Three-Dimensional Water Waves, Studies in Applied Mathematics, 97, 149–166, 1996.
- b. Peterson, P., Soomere, T., Engelbrecht, J., and van Groesen, E.: Soliton interaction as a possible model for extreme waves in shallow water, Nonlinear Processes in Geophysics, 10, 503–510, 2003.
- 3) We have extended the article's title;
- 4) We have improved Fig. 9 and Fig. 10.

Other changes:

- 1) We have corrected some typos and improved the grammar;
- 2) We have added references to the following paper:
  - Kalogirou, A. and Bokhove, O.: Mathematical and numerical modelling of wave impact on wave-energy buoys, in: Proceedings of the ASME 2016 35th International Conference on Ocean, Offshore and Arctic Engineering, 2016.
- 3) We have added Dr. Anna Kalogirou as a co-author following her initial and continuing contributions.

All of the changes summarized above are highlighted in the following revised manuscript.

[revised manuscript text omitted]
^{im}_{-\sim} \varphi^{im}_{-\sim}(t), \qquad \qquad Q^{\tau} = Q^{im}_{-\sim} \psi^{im}_{-\sim}(t). \tag{53}$$

The linear basis functions  $\varphi^i$  and  $\psi^i \varphi^m$  and  $\psi^m$  are continuous within each time interval, but admit discontinuities at the interface between two time slots. Therefore, to discretize discretize Eq. (52), the notion of jumps [[.]] and average averages  $\{\{.\}\}^{\beta}_{\alpha}$  for a time dependent time-dependent function d(t) must be introduced (Gagarina et al., 2016):

$$[[d]]|_{\underline{t_n} t^n_{\sim}} = d^{n,-} - d^{n,+}, \quad \text{and} \quad \{\{d\}\}^{\beta}_{\alpha}|_{\underline{t_n} t^n_{\sim}} = \alpha d^{n,-} + \beta d^{n,+}.$$
(54)

5 The coefficients  $\alpha$  and  $\beta$  are real numbers defined such that  $\alpha + \beta = 1$  and  $\alpha, \beta \ge 0$ . The notation  $d^{n,\pm}$  denotes the left and right traces of d(t) at time  $t_n t_{\alpha}^n$ , that is

$$d^{n,\pm} = \lim_{\epsilon \to 0} d(t_{\underline{n}_{\sim}}^{n} \pm \epsilon).$$
(55)

Discretization Discretisation of the variational principle Eq. (52) then yields (Gagarina et al., 2016)

$$\delta \mathcal{L}^{\tau}(P^{\tau}, Q^{\tau}, t) = \delta \left[ \sum_{n=0}^{N-1} \int \frac{t_{n+1} t^{n+1}}{\frac{t_n}{t_n} t_{\infty}^n} \left( P^{\tau} \frac{dQ^{\tau}}{dt} - H(Q^{\tau}, P^{\tau}) \right) dt - \sum_{n=-1}^{N-1} [[Q^{\tau}]] \{\{P^{\tau}\}\}_{\alpha}^{\beta} \big|_{\underline{t_{n+1}} t_{\infty}^{n+1}} \right], \tag{56}$$

10 where  $\mathbb{N}_{\mathcal{N}}$  is the number of finite time intervals  $[t_n, t_{n+1}]$   $[t_n^n, t_{n+1}]$  that divide the time domain [0, T]. Gagarina et al. (2016) showed that to obain a second-order Stormer-Verlet Störmer-Verlet scheme, P and Q must be discretized with a discretized with a discretized with trapezoidal and mid-point rules respectively, that is:

$$P^{\tau} = \frac{t^{n+1} - t}{\Delta t} P^{n,+} + \frac{t - t^n}{\Delta t} P^{n+1,-},$$
(57)

$$Q^{\tau} = \frac{2(t-t^{n})}{\Delta t}Q^{n+1/2} + \frac{t^{n}+t^{n+1}-2t}{\Delta t}Q^{n}, + \frac{t^{n}+t^{n}+t^{n+1}-2t}{\Delta t}Q^{n}, + \frac{t^{n}+t^{n}+t^{n}-2t}{\Delta t}Q^{n}, + \frac{t^{n}+t^{n}+t^{n}+t^{n}+t^{n}+t^{n}+t^{n}+t^{n}+t^{n}+t^{n}+t^{n}+t^{n}+t^{n}+t^{n}+t^{n}+t^{n}+t^{n}+t^{n}+t^{n}+t^{n}+t^{n}+t^{n}+t^{n}+t^{n}+t^{n}+t^{n}+t^{n}+t^{n}+t^{n}+t^{n}+t^{n}+t^{n}+t^{n}+t^{n}+t^{n}+t^{n}+t^{n}+t^{n}+t^{n}+t^{n}+t^{n}+t^{n}+t^{n}+t^{n}+t^{n}+t^{n}+t^{n}+t^{n}+t^{n}+t^{n}+t^{n}+t^{n}+t^{n}+t^{n}+t^{n}+t^{n}+t^{n}+t^{n}+t^{n}+t^{n}+t^{n}+t^{n}+t^{n}+t^{n}+t^{n}+t^{n}+t^{n}+t^{n}+t^{n}+t^{n}+t^{n}+t^{n}+t^{n}+t^{n}+t^{n}+t^{n}+t^{n}+t^{n}+t^{n}+t^{n}+t^{n}+t^{n}+t^{n}+t^{n}+t^{n}+t^{n}+t^{n}+t^{n}+t^{n}+t^{n}+t^{n}+t^{n}+t^{n}+t^{n}+t^{n}+t^{n}+t^{n}+t^{n}+t^{n}+t^{n}+t^{n}+t^{n}+t^{n}+t^{n}+t^{n}+t^{n}+t^{n}+t^{n}+t^{n}+t^{n}+t^{n}+t^{n}+t^{n}+t^{n}+t^{n}+t^{n}+t^{n}+t^{n}+t^{n}+t^{n}+t^{n}+t^{n}+t^{n}+t^{n}+t^{n}+t^{n}+t^{n}+t^{n}+t^{n}+t^{n}+t^{n}+t^{n}+t^{n}+t^{n}+t^{n}+t^{n}+t^{n}+t^{n}+t^{n}+t^{n}$$

[revised manuscript text omitted]
 = 0 \quad \text{for} \quad \kappa \ge 1\\ \varphi_w > 0 \quad \text{for} \quad \kappa < 1. \end{cases}$$
(65)

10 Predictions (64) and (65) are now being checked numerically were checked numerically as discussed next.

**5.3.1 Regular reflection**

/

Figure 9 shows numerical results and expectations for the specific predictions for the case where  $\kappa = 1.12 \ge 1$ . The wall makes an angle of 30° with the direction of propagation of the initial solitary wave, hence  $\varphi_i$  equals 30°. On  $\varphi_i = 30^\circ$ . In the bottomright plot of Fig. 9, one can measure there is an angle of 60° between the reflected and stem waves which means that the angle

15  $\varphi_r$  between the reflected wave and the line perpendicular to the wall is equal to  $30^\circ$ , that is, equal to  $\varphi_i$ . This observation holds at any time and therefore the expectations (64) for the reflected waves are satisfied in the case of regular reflection. The stem wave propagates along the wall without increasing in length, and therefore no angle can be measured between the stem wave and the wall  $\div i.e., \varphi_w = 0$ , as predicted in (65) for regular reflection. These results, together with Fig. 8 for the amplification of the stem wave, confirm modified-Miles' theory in the case  $\kappa \ge 1$ , for both the reflected and stem waves.

**20 5.3.2 Mach reflection**

Figure 10 shows numerical results and schematic expectations for the propagation of the reflected and stem wave for κ = 0.58 < 1. In the bottom right plot, one can first measure bottom-right sub-figure, the angle between the incident and reflected waves can be measured, as represented in the top right scheme, top-right sub-figure, in order to check that φr is larger than φi. The total angle φr + φi measures 70°, with the initial incident angle set to φi = 30°. Therefore, φr measures is 40° and, which is indeed larger than φi, which corresponds to the 
[revised manuscript text omitted]